# GENERATING VIDEOS WITH DYNAMICS-AWARE IMPLICIT GENERATIVE ADVERSARIAL NETWORKS

**Sihyun Yu**[*,1]**, Jihoon Tack**[*,1]**, Sangwoo Mo**[*,1]**,**
**Hyunsu Kim**[2]**, Junho Kim**[2]**, Jung-Woo Ha**[2]**, Jinwoo Shin**[1]
[1]Korea Advanced Institute of Science and Technology (KAIST), [2]NAVER AI Lab
{sihyun.yu, jihoontack, swmo, jinwoos}@kaist.ac.kr,
{hyunsu1125.kim, jhkim.ai, jungwoo.ha}@navercorp.com

## ABSTRACT

In the deep learning era, long video generation of high-quality still remains challenging due to the spatio-temporal complexity and continuity of videos. Existing prior works have attempted to model video distribution by representing videos as 3D grids of RGB values, which impedes the scale of generated videos and neglects continuous dynamics. In this paper, we found that the recent emerging paradigm of implicit neural representations (INRs) that encodes a continuous signal into a parameterized neural network effectively mitigates the issue. By utilizing INRs of video, we propose *dynamics-aware implicit generative adversarial network* (DI-GAN), a novel generative adversarial network for video generation. Specifically, we introduce (a) an INR-based video generator that improves the motion dynamics by manipulating the space and time coordinates differently and (b) a motion discriminator that efficiently identifies the unnatural motions without observing the entire long frame sequences. We demonstrate the superiority of DIGAN under various datasets, along with multiple intriguing properties, *e.g.*, long video synthesis, video extrapolation, and non-autoregressive video generation. For example, DIGAN improves the previous state-of-the-art FVD score on UCF-101 by 30.7% and can be trained on 128 frame videos of 128×128 resolution, 80 frames longer than the 48 frames of the previous state-of-the-art method.[1][*]

## 1 INTRODUCTION

Deep generative models have successfully synthesized realistic samples on various domains, including image (Brock et al., 2019; Karras et al., 2020b; 2021; Dhariwal & Nichol, 2021), text (Adiwardana et al., 2020; Brown et al., 2020), and audio (Dhariwal et al., 2020; Lakhotia et al., 2021). Recently, video generation has emerged as the next challenge of deep generative models, and thus a long line of work has been proposed to learn the video distribution (Vondrick et al., 2016; Kalchbrenner et al., 2017; Saito et al., 2017; 2020; Tulyakov et al., 2018; Acharya et al., 2018; Clark et al., 2019; Weissenborn et al., 2020; Rakhimov et al., 2020; Tian et al., 2021; Yan et al., 2021).

Despite their significant efforts, a substantial gap still exists from large-scale real-world videos. The difficulty of video generation mainly stems from the complexity of video signals; they are continuously correlated across spatio-temporal directions. Specifically, most prior works interpret the video as a 3D grid of RGB values, *i.e.*, a sequence of 2D images, and model them with discrete decoders such as convolutional (Tian et al., 2021) or autoregressive (Yan et al., 2021) networks. However, such discrete modeling limits the scalability of generated videos due to the cubic complexity (Saito et al., 2020) and ignores the inherent continuous temporal dynamics (Gordon & Parde, 2021).

Meanwhile, implicit neural representations (INRs; Sitzmann et al. (2020); Tancik et al. (2020)) have emerged as a new paradigm for representing continuous signals. INR encodes a signal into a neural network that maps input coordinates to corresponding signal values, *e.g.*, 2D coordinates of images to RGB values. Consequently, INR amortizes the signal values of arbitrary coordinates into a compact neural representation instead of discrete grid-wise signal values, requiring a large memory

---

[1]Videos are available at the project site https://sihyun-yu.github.io/digan/.
[*]Equal contribution.

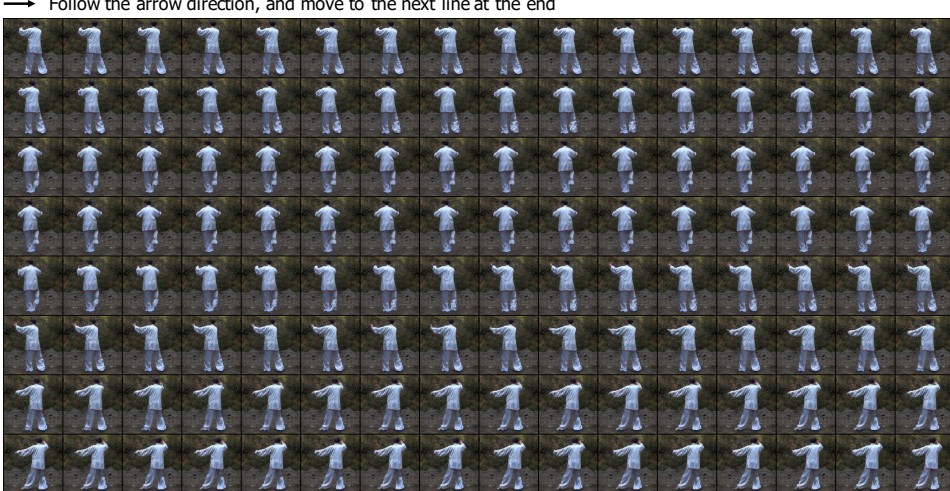

Figure 1: 128 frame video of 128×128 resolution generated by DIGAN on the Tai-Chi-HD dataset. DIGAN can train these videos with 4 NVIDIA V100 GPUs, while the prior state-of-the-art method, DVD-GAN, uses more than 32 (up to 512) TPUs for training 48 frame videos of the same resolution.

proportional to coordinate dimension and resolution. In this respect, INRs have shown to be highly effective at modeling complex signals such as 3D scenes (Mildenhall et al., 2020; Li et al., 2021a). Furthermore, INR has intriguing properties from its compactness and continuity, *e.g.*, reduced data memory (Dupont et al., 2021a) and upsampling to arbitrary resolution (Chen et al., 2021b).

Several works utilize INRs for generative modeling, *i.e.*, samples are generated through INRs (Chan et al., 2021; Dupont et al., 2021b; Kosiorek et al., 2021). In particular, Skorokhodov et al. (2021a) and Anokhin et al. (2021) exposed that INR-based image generative adversarial networks (GANs; Goodfellow et al. (2014)), which generate images as INRs, show impressive generation performance. Interestingly, they further merit various advantages of INRs, *e.g.*, natural inter- and extra-polation, anycost inference (*i.e.*, control the trade-off of quality and cost), and parallel computation, which needs a non-trivial modification to apply under other generative model architectures.

Inspired by this, we aim to design an INR-based (or implicit) video generation model by interpreting videos as continuous signals. This alternative view is surprisingly effective as INRs compactly encode the videos without 3D grids and naturally model the continuous spatio-temporal dynamics. While naïvely applying INRs for videos is already fairly effective, we found that a careful design of separately manipulating space and time significantly improves the video generation.

**Contribution.** We introduce *dynamics-aware implicit generative adversarial network* (DIGAN), a novel INR-based GAN architecture for video generation. Our idea is two-fold (see Figure 2):

- *Generator:* We propose an INR-based video generator that decomposes the motion and content (image) features, and incorporates the temporal dynamics into the motion features.[2] To be specific, our generator encourages the temporal coherency of videos by regulating the variations of motion features with a smaller temporal frequency and enhancing the expressive power of motions with an extra non-linear mapping. Moreover, our generator can create videos with diverse motions sharing the initial frame by conditioning a random motion vector to the content vector.

- *Discriminator:* We propose a motion discriminator that efficiently detects unnatural motions from a pair of images (and their time difference) instead of a long sequence of images. Specifically, DIGAN utilizes a 2D convolutional network for the motion discriminator, unlike prior works that utilize computationally heavier 3D convolutional networks to handle the entire video at once. Such an efficient discriminating scheme is possible since INRs of videos can non-autoregressively synthesize highly correlated frames at arbitrary times.

We demonstrate the superiority of DIGAN on various datasets, including UCF-101 (Soomro et al., 2012), Tai-Chi-HD (Siarohin et al., 2019), Sky Time-lapse (Xiong et al., 2018), and a food class subset of Kinetics-600 (Carreira et al., 2018) datasets, *e.g.*, it improves the state-of-the-art results

---

[2]We remark that this decomposition can be naturally done under the INR framework, and one can effectively extend the implicit image GANs for video generation by adding a motion generator.

of Fréchet video distance (FVD; Unterthiner et al. (2018), lower is better) on UCF-101 from 833 to 577 (+30.7%). Furthermore, DIGAN appreciates various intriguing properties, including,

- *Long video generation*: synthesize long videos of high-resolution without demanding resources on training, *e.g.*, 128 frame video of 128×128 resolution (Figure 1)
- *Time interpolation and extrapolation*: fill in the interim frames to make videos transit smoother, and synthesize the out-of-frame videos (Figure 4, Table 2)
- *Non-autoregressive generation*: generate arbitrary time frames, *e.g.*, previous scenes (Figure 5), and fast inference via parallel computing of multiple frames (Table 3)
- *Diverse motion sampling*: sample diverse motions from the shared initial frames (Figure 6)
- *Space interpolation and extrapolation*: upsample the resolution of videos (Figure 7, Table 4) and create zoomed-out videos while preserving temporal consistency (Figure 8)

To the best of our knowledge, we are the first to leverage INRs for video generation. We hope that our work would guide new intriguing directions for both video generation and INRs in the future.

## 2 RELATED WORK

**Image generation.** Image generation has achieved remarkable progress, with the advance of various techniques, including generative adversarial networks (GANs; Goodfellow et al. (2014)), autoregressive models (Ramesh et al., 2021), and diffusion models (Dhariwal & Nichol, 2021). In particular, GANs have been considered as one of the common practices for image generation, due to the fast inference at synthesizing high-resolution images (Brock et al., 2019; Karras et al., 2020b; 2021). Inspired by recent GAN architectures (Karras et al., 2020b; Skorokhodov et al., 2021a) and training techniques, (Zhao et al., 2020; Karras et al., 2020a), we extend these methods for video generation.

**Video generation.** Following the success of GANs on images, most prior works on video generation considered the temporal extension of image GANs (Vondrick et al., 2016; Saito et al., 2017; 2020; Tulyakov et al., 2018; Acharya et al., 2018; Clark et al., 2019; Yushchenko et al., 2019; Kahembwe & Ramamoorthy, 2020; Gordon & Parde, 2021; Tian et al., 2021; Fox et al., 2021; Munoz et al., 2021). Another line of works (Kalchbrenner et al., 2017; Weissenborn et al., 2020; Rakhimov et al., 2020; Yan et al., 2021) train autoregressive models over pixels or discretized embeddings. Despite their significant achievement, a large gap still exists between generated results and large-scale real-world videos. We aim to move towards longer video generation by exploiting the power of implicit neural representations. We provide more discussion on related fields in Appendix C.

**Implicit neural representations.** Implicit neural representations (INRs) have recently gained considerable attention, observing that high-frequency sinusoidal activations significantly improve continuous signal modeling (Sitzmann et al., 2020; Tancik et al., 2020). In particular, INR has proven its efficacy in modeling complex signals such as static (Chen et al., 2021b; Park et al., 2021; Martin-Brualla et al., 2021) and dynamic (Li et al., 2021a;b; Pumarola et al., 2021; Xian et al., 2021) 3D scenes, and 2D videos (Chen et al., 2021a). Unlike prior works on INRs focusing on modeling a *single* signal, we learn a generative model over 2D videos. Moreover, while prior works focus more on rendering visually appealing 3D scenes, we aim to learn the diverse motion dynamics of videos.

**Generative models with INRs.** Following the success of single signals, several works utilize INRs for generative modeling. One line of works synthesizes the INR weights correspond to the signals via hypernetwork (Skorokhodov et al., 2021a; Anokhin et al., 2021; Chan et al., 2021; Dupont et al., 2021b). The other line of works controls the generated signals via input condition, *i.e.*, concatenate the latent code corresponding to the signal to the input coordinates (Schwarz et al., 2020; Kosiorek et al., 2021). In particular, Skorokhodov et al. (2021a) and Anokhin et al. (2021) have demonstrated the effectiveness of INR-based generative models for image synthesis. We extend the applicability of INR-based GANs for video generation by incorporating temporal dynamics.

## 3 DYNAMICS-AWARE IMPLICIT GENERATIVE ADVERSARIAL NETWORK

The goal of video generation is to learn the model distribution $p_G(\mathbf{v})$ to match with the data distribution $p_{\text{data}}(\mathbf{v})$, where a video $\mathbf{v}$ is defined as a continuous function of images $\mathbf{i}_t$ for time $t \in \mathbb{R}$, and an image $\mathbf{i}_t$ is a function of spatial coordinates $(x, y) \in \mathbb{R}^2$. To this end, most prior works considered

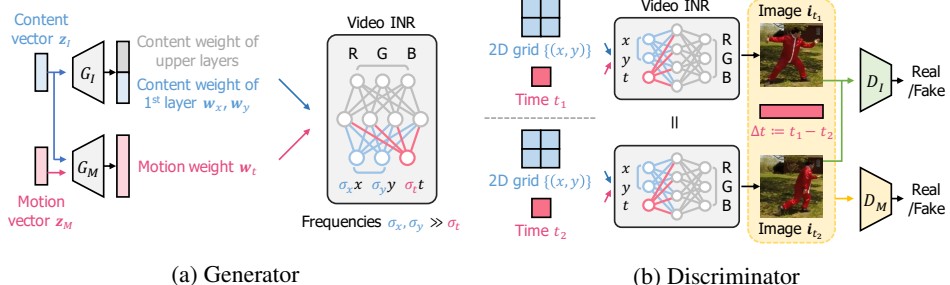

(a) Generator        (b) Discriminator

Figure 2: Illustration of the (a) generator and (b) discriminator of DIGAN. The generator creates a video INR weight from random content and motion vectors, which produces an image that corresponds to the input 2D grids $\{(x, y)\}$ and time $t$. Two discriminators determine the reality of each image and motion (from a pair of images and their time difference), respectively.

videos as the discretized sequence of images, *i.e.*, 3D grid of RGB values of size $H \times W \times T$, and generated videos with discrete decoders such as convolutional (Tian et al., 2021) or autoregressive (Yan et al., 2021) networks. However, the discretization limits video generation hard to scale due to the cubic complexity of generated videos and ignores the continuous dynamics.

Our key idea is to directly model the videos as continuous signals using implicit neural representations (INRs; Sitzmann et al. (2020); Tancik et al. (2020)). Specifically, we propose *dynamics-aware implicit generative adversarial network* (DIGAN), an INR-based generative adversarial network (GAN; Goodfellow et al. (2014)) for video generation. Inspired by the success of INR-based GANs for image synthesis (Skorokhodov et al., 2021a; Anokhin et al., 2021), DIGAN extends the implicit image GANs for video generation by incorporating temporal dynamics. We briefly review INRs and INR-based GANs in Section 3.1 and then introduce our method DIGAN in Section 3.2.

### 3.1 GENERATIVE MODELING WITH IMPLICIT NEURAL REPRESENTATIONS

Consider a signal $\mathbf{v}(\cdot) : \mathbb{R}^m \to \mathbb{R}^n$ of a coordinate mapping to the corresponding signal values, *e.g.*, videos as $\mathbf{v}(x, y, t) = (r, g, b)$ with $m, n = 3$, where $(x, y, t)$ are space-time coordinates and $(r, g, b)$ are RGB values. Without loss of generality, we assume that the range of coordinates for signals are $[0, 1]$, *e.g.*, $[0, 1]^3$ for videos. INR aims to directly model the signal with a neural network $\mathbf{v}(\cdot; \phi) : \mathbb{R}^m \to \mathbb{R}^n$ parametrized by $\phi$, *e.g.*, using a multi-layer perceptron (MLP). Recently, Tancik et al. (2020) and Sitzmann et al. (2020) found that sinusoidal activations with a high-frequency input $\sigma x$, *i.e.*, $\sin(\sigma x)$ for $\sigma \gg 1$, significantly improve the modeling of complex signals like 3D scenes (Mildenhall et al., 2020) when applied on the first layer (or entire layers). Here, one can decode the INR $\mathbf{v}(\cdot; \phi)$ as a standard discrete grid interpretation of signals by computing the values of predefined grid of input coordinates, *e.g.*, $\{(\frac{i-1}{H-1}, \frac{j-1}{W-1}, \frac{k-1}{T-1})\} \subset [0, 1]^3$ for videos of size $H \times W \times T$.

Leveraging the power of INRs, several works utilize them for generative models, *i.e.*, the generator $G(\cdot)$ maps a latent $\mathbf{z} \sim p(\mathbf{z})$ from a given prior distribution $p(\mathbf{z})$ to an INR parameter $\phi = G(\mathbf{z})$ that corresponds to the generated signal $\mathbf{v}(\cdot; \phi)$, *e.g.*, Chan et al. (2021). Remark that INR-based generative models synthesize the *function* of coordinates, unlike previous approaches that directly predict the outputs of such a function, *e.g.*, 2D grid of RGB values for image synthesis. Hence, INR-based generative models only need to synthesize a fixed size parameter $\phi$, which reduces the complexity of the generated outputs for complicated signals. It is especially important for higher-dimensional signals than 2D images, *e.g.*, video synthesis of grids require cubic complexity.

Specifically, Skorokhodov et al. (2021a) and Anokhin et al. (2021) propose GAN frameworks for training INR-based (or implicit) model for image generation, *i.e.*, joint training of the discriminator $D(\cdot)$ to distinguish the real and generated samples, while the generator $G(\cdot)$ aims to fool the discriminator.[3] In particular, they present a new generator to synthesize weights of INRs and employ conventional convolutional GAN discriminator architectures to identify the fake images decoded from INRs. Remarkably, they have shown comparable results with state-of-the-art GANs while meriting intriguing properties, *e.g.*, inter- and extra-polation and fast inference (Skorokhodov et al., 2021a).

---

[3]By gradually improving their abilities from two-player game, the generator converges to the data distribution, *i.e.*, $p_G \approx p_{\text{data}}$, where $p_G$ is a distribution of $G(\mathbf{z})$ for $\mathbf{z} \sim p(\mathbf{z})$ (Mescheder et al., 2018).

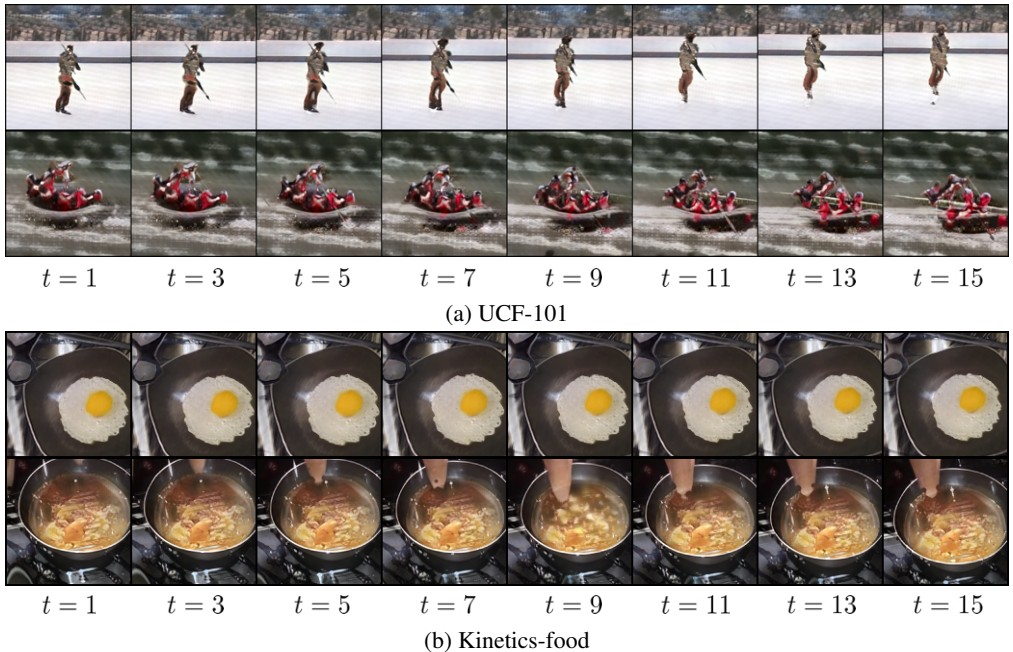

$t = 1 \qquad t = 3 \qquad t = 5 \qquad t = 7 \qquad t = 9 \qquad t = 11 \qquad t = 13 \qquad t = 15$

(a) UCF-101

$t = 1 \qquad t = 3 \qquad t = 5 \qquad t = 7 \qquad t = 9 \qquad t = 11 \qquad t = 13 \qquad t = 15$

(b) Kinetics-food

Figure 3: Generated video results of DIGAN on UCF-101 and Kinetics-food datasets.

## 3.2 Incorporating temporal dynamics for implicit GANs

Our method, DIGAN, is an INR-based video GAN that incorporates the temporal dynamics into the INRs. Recall that the video INRs only differ from the image INRs by an extra time coordinate, *i.e.*, an input becomes a 3D coordinate $(x, y, t)$ from a 2D coordinate $(x, y)$; hence, one can utilize the implicit image GANs for video generation by only expanding the input dimension of INRs by one (for the time coordinate). However, we found that a careful generator and discriminator design notably improves the generation quality and training efficiency. We provide the overall illustration of DIGAN in Figure 2, and explain the details of DIGAN in the remaining section.

**Generator.** A naïve extension of implicit image GANs for synthesizing video INRs $\mathbf{v}(x, y, t; \phi)$ is to utilize them intactly, only modifying the first layer of INRs to handle the extra time coordinate (we assume that the INRs follow the MLP structure). This approach synthesizes the entire INR parameter $\phi$ at once, which overlooks the difference in space and time of videos, *e.g.*, the smooth change of frames over the time direction. To alleviate this issue, we first notice that the spatial and temporal terms of video INRs can be easily decomposed. Specifically, the output of the first layer (without sinusoidal activations) of video INRs can be interpreted as $\sigma_x \boldsymbol{w}_x x + \sigma_y \boldsymbol{w}_y y + \sigma_t \boldsymbol{w}_t t + \boldsymbol{b}$, where $\boldsymbol{w}_x, \boldsymbol{w}_y, \boldsymbol{w}_t, \boldsymbol{b}$ are weights and biases of the first layer and $\sigma_x, \sigma_y, \sigma_t > 0$ are frequencies of coordinates $x, y, t$. Here, note that only the term $\sigma_t \boldsymbol{w}_t t$ differs from the image INRs, so the outputs (of the first layer) of video INRs can be viewed as a continuous trajectory over time $t$ from the initial frame (at $t = 0$) determined by the content parameter $\phi_I \coloneqq \phi \setminus \{\boldsymbol{w}_t\}$. Inspired by this space-time decomposition view, we incorporate the temporal dynamics into the motion parameter $\boldsymbol{w}_t$.

To be specific, we propose three components to improve the motion parameter $\boldsymbol{w}_t$ considering the temporal behaviors. First, we use a smaller time-frequency $\sigma_t$ than space-frequencies $\sigma_x, \sigma_y$ since the video frames change relatively slowly over time compared to the spatial variations covering diverse objects in images. It encourages the videos to be coherent over time. Second, we sample a latent vector for motion diversity $\boldsymbol{z}_M \sim p_M(\boldsymbol{z}_M)$ in addition to the original content (image) latent vector $\boldsymbol{z}_I \sim p_I(\boldsymbol{z}_I)$. Here, the content parameter $\phi_I = G_I(\boldsymbol{z}_I)$ is generated as the prior implicit image GANs, but the motion parameter $\boldsymbol{w}_t = G_M(\boldsymbol{z}_I, \boldsymbol{z}_M)$ is conditioned on both content and motion vectors; since possible motions of a video often depend on the content. Finally, we apply a non-linear mapping $f_M(\cdot)$ on top of the motion features at time $t$ to give more freedoms to motions, *i.e.*, $f_M(\boldsymbol{w}_t t)$. These simple modifications further improves the generation quality (Section 4.1) while creates diverse motions (Section 4.2).

Table 1: IS, FVD, and KVD values of video generation models on (a) UCF-101, (b) Sky, (c) TaiChi, and (d) Kinetics-food datasets. ↑ and ↓ imply higher and lower values are better, respectively. Subscripts denote standard deviations, and bolds indicate the best results. "Train split" and "Train+test split" denote whether the model is trained with the train split (following the setup in Saito et al. (2020)) or with the full dataset (following the setup in Tian et al. (2021)), respectively.

(a) UCF-101

| Method | IS (↑) | FVD (↓) |
|---|---|---|
| *Train split* | | |
| VGAN | $8.31_{\pm.09}$ | - |
| TGAN | $11.85_{\pm.07}$ | - |
| MoCoGAN | $12.42_{\pm.07}$ | - |
| ProgressiveVGAN | $14.56_{\pm.05}$ | - |
| LDVD-GAN | $22.91_{\pm.19}$ | - |
| VideoGPT | $24.69_{\pm.30}$ | - |
| TGANv2 | $28.87_{\pm.67}$ | $1209_{\pm28}$ |
| DIGAN (ours) | $\mathbf{29.71_{\pm.53}}$ | $\mathbf{655_{\pm22}}$ |
| *Train+test split* | | |
| DVD-GAN | $27.38_{\pm.53}$ | - |
| MoCoGAN-HD | 32.36 | 838 |
| DIGAN (ours) | $\mathbf{32.70_{\pm.35}}$ | $\mathbf{577_{\pm21}}$ |

(b) Sky

| Method | FVD (↓) | KVD (↓) |
|---|---|---|
| MoCoGAN-HD | $183.6_{\pm5.2}$ | $13.9_{\pm0.7}$ |
| DIGAN (ours) | $\mathbf{114.6_{\pm4.3}}$ | $\mathbf{6.8_{\pm0.5}}$ |

(c) TaiChi

| Method | FVD (↓) | KVD (↓) |
|---|---|---|
| MoCoGAN-HD | $144.7_{\pm6.0}$ | $25.4_{\pm1.9}$ |
| DIGAN (ours) | $\mathbf{128.1_{\pm4.9}}$ | $\mathbf{20.6_{\pm1.1}}$ |

(d) Kinetics-food

| Method | FVD (↓) | KVD (↓) |
|---|---|---|
| MoCoGAN-HD | $430.4_{\pm29.9}$ | $276.0_{\pm50.7}$ |
| DIGAN (ours) | $\mathbf{313.3_{\pm36.9}}$ | $\mathbf{183.0_{\pm40.3}}$ |

**Discriminator.** As the generated videos should be natural in both images and motions, prior works on video GANs are commonly equipped with two discriminators: an image discriminator $D_I$ and a motion (or video) discriminator $D_M$ (Clark et al., 2019; Tian et al., 2021).[4] For the image discriminator, one can utilize the well-known 2D convolutional architectures from the image GANs. However, the motion discriminator needs an additional design, *e.g.*, 3D convolutional networks, where inputs are a sequence of images (*e.g.*, the entire videos). The 3D convolutional motion discriminators are the main computational bottleneck of video GANs, and designing an efficient video discriminator has been widely investigated (Saito et al., 2020; Kahembwe & Ramamoorthy, 2020).

Instead of the computationally expensive 3D architectures, we propose an efficient 2D convolutional video discriminator. Here, we emphasize that video INRs can efficiently generate two frames of arbitrary times $t_1, t_2$ unlike autoregressive models which require generating the entire sequence. Utilizing this unique property of INRs, we adopt the image discriminator to distinguish the *triplet* consists of a pair of images and their time difference $(\mathbf{i}_{t_1}, \mathbf{i}_{t_2}, \Delta t)$ for $\Delta t := |t_1 - t_2|$, by expanding the input channel from 3 to 7. Intriguingly, this discriminator can learn the dynamics without observing the whole sequence (Section 4.1). Moreover, we note that the two frames of INRs are highly correlated due to their continuity; the discriminator can focus on the artifacts in the motion.

## 4 EXPERIMENTS

We present the setups and main video generation results in Section 4.1. We then exhibit the intriguing properties of DIGAN in Section 4.2. Finally, we conduct ablation studies in Section 4.3.

### 4.1 VIDEO GENERATION

**Model.** We implement DIGAN upon the INR-GAN (Skorokhodov et al., 2021a) architecture, an INR-based GAN for image generation. The content generator $G_I$ is identical to the INR-GAN generator, but the motion generator $G_M$ is added. We set the spatial frequencies $\sigma_x = \sigma_y = \sqrt{10}$ following the original configuration of INR-GAN, but we found that using a smaller value for the temporal frequency $\sigma_t$ performs better; we use $\sigma_t = 0.25$ for all experiments ff We use the same discriminator of INR-GAN for both image and motion discriminators but only differ for the input channels: 3 and 7. See Appendix A.1 for details.

---

[4]While a video discriminator alone can learn the video distribution (Vondrick et al., 2016), recent works often use both discriminators since such separation empirically performs better (Tulyakov et al., 2018).

Table 2: FVD values of generated videos inter- and extra-polated over time. All models are trained on 16 frame videos of 128×128 resolution. The videos are interpolated to 64 frames (*i.e.*, 4× finer) and extrapolated 16 more frames. We measure FVD with 512 samples for Sky, since the test data size becomes less than 2,048.

| | Interpolation | | | Extrapolation | | |
|---|---|---|---|---|---|---|
| Method | Sky | TaiChi | Kinetics-food | Sky | TaiChi | Kinetics-food |
| MoCoGAN-HD | 402.2±18.9 | 249.0±12.7 | 1029.8±28.4 | 303.2±4.3 | 337.8±3.7 | 877.8±22.6 |
| DIGAN (ours) | **324.2±20.5** | **241.6±7.5** | **722.2±20.1** | **224.3±6.2** | **289.3±15.6** | **693.7±14.1** |

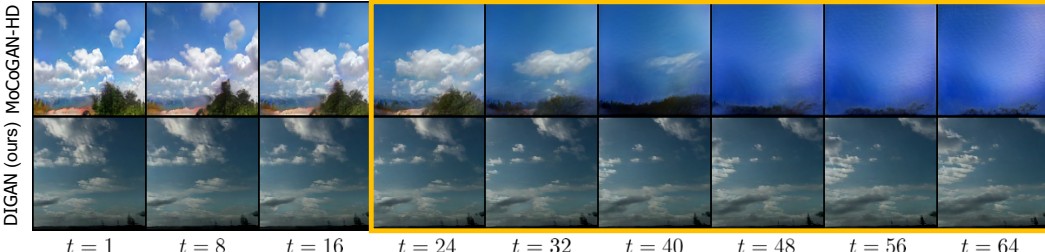

Figure 4: Generated videos of MoCoGAN-HD and DIGAN, trained on 16 frame videos of 128×128 resolution on the Sky dataset. Yellow box indicates the extrapolated results until 64 frames.

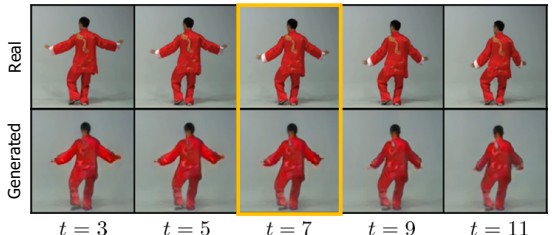

Figure 5: Forward and backward prediction results of DIGAN. Yellow box indicates the given frame.

Table 3: Time (sec) for generating a 128×128 video for VideoGPT, MoCoGAN-HD, and DIGAN. Bolds indicate the best results.

| | Video length | | |
|---|---|---|---|
| Method | 16 | 32 | 64 |
| VideoGPT | ∼40 | ∼80 | ∼170 |
| MoCoGAN-HD | 0.154 | 0.303 | 0.612 |
| DIGAN (ours) | **0.069** | **0.132** | **0.260** |

**Datasets and evaluation.** We conduct the experiments on UCF-101 (Soomro et al., 2012), Tai-Chi-HD (TaiChi; Siarohin et al. (2019)), Sky Time-lapse (Sky; Xiong et al. (2018)), and a food class subset of Kinetics-600 (Kinetics-food; Carreira et al. (2018)) datasets. All models are trained on 16 frame videos of 128×128 resolution unless otherwise specified. Specifically, we use the consecutive 16 frames for UCF-101, Sky, and Kinetics-food, but stride 4 (*i.e.*, skip 3 frames after the chosen frame) for TaiChi to make motion more dynamic. Following prior works, we report the Inception score (IS; Salimans et al. (2016)), Fréchet video distance (FVD; Unterthiner et al. (2018)), and kernel video distance (KVD; Unterthiner et al. (2018)). We average 10 runs for main results and 5 runs for analysis with standard deviations. See Appendix A.2 for more details.

**Baselines.** We mainly compare DIGAN with prior works on UCF-101, a commonly used benchmark dataset for video generation. Specifically, we consider VGAN (Vondrick et al., 2016), TGAN (Saito et al., 2017), MoCoGAN (Tulyakov et al., 2018), ProgressiveVGAN (Acharya et al., 2018), LDVD-GAN (Kahembwe & Ramamoorthy, 2020), VideoGPT (Yan et al., 2021), TGANv2 (Saito et al., 2020), DVD-GAN (Clark et al., 2019), and MoCoGAN-HD (Tian et al., 2021), where the values are collected from the references. For other experiments, we compare DIGAN with the state-of-the-art method, MoCoGAN-HD, tested using the official code. See Appendix B for details.

**Main results.** Figure 3 and Table 1 present the qualitative and quantitative video generation results of DIGAN, respectively. DIGAN can model various video distributions, including unimodal videos like Sky and TaiChi and multimodal videos like UCF-101 and Kinetics-food. In particular, Figure 3 presents that DIGAN produces reasonably good videos for challenging multimodal videos. Also, Table 1 exhibits that DIGAN significantly outperforms the prior work on all datasets, *e.g.*, improves the FVD of MoCoGAN-HD from 833 to 577 (+30.7%) on UCF-101. We remark that the Fréchet inception distance (FID; Heusel et al. (2017)), a metric for image quality, of DIGAN is similar to the MoCoGAN-HD. Thus, the FVD gains of DIGAN come from better dynamics modeling.

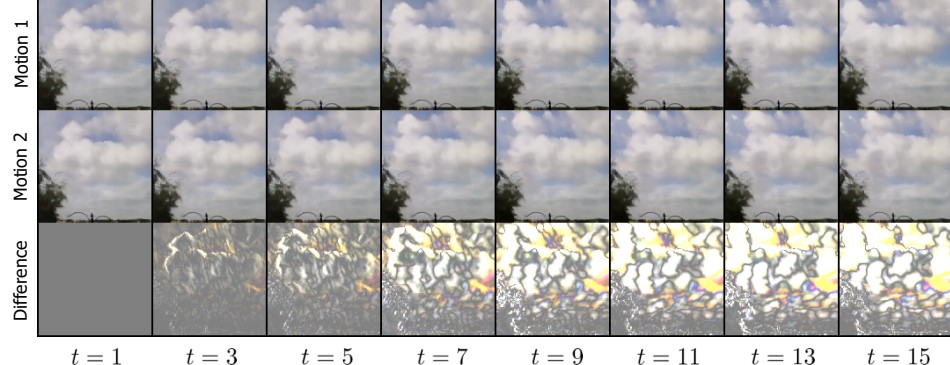

Figure 6: Videos sampled from two random motion vectors. The first two rows are generated videos, and the third row is the pixel difference between the two videos (yellow implies more differences).

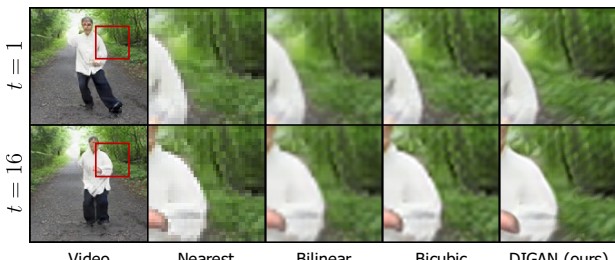

Figure 7: Videos upsampled from 128×128 to 512×512 resolution (4× larger) on TaiChi dataset.

Table 4: FVD values of videos upsampled from 128×128 to 256×256 resolution (2× larger) on TaiChi dataset.

| Method | FVD ($\downarrow$) |
|---|---|
| Nearest | $180.6_{\pm 5.1}$ |
| Bilinear | $236.7_{\pm 6.7}$ |
| Bicubic | $175.9_{\pm 5.4}$ |
| DIGAN (ours) | $\mathbf{156.7}_{\pm \mathbf{6.2}}$ |

## 4.2 INTRIGUING PROPERTIES

**Long video generation.** The primary advantage of DIGAN is an effective generation of long and high-quality videos, leveraging the compact representations of INRs. We also remark that DIGAN can be efficiently trained on the long videos, as the motion discriminator of DIGAN only handles a pair of images instead of long image sequences. To verify the efficacy of DIGAN, we train a model using 128 frame videos of 128×128 resolution from the TaiChi dataset. Figure 1 shows that DIGAN produces a long and natural motion with reasonable visual quality. To our best knowledge, we are the first to report 128 frame videos of this quality. See Appendix F for more results.

**Time interpolation and extrapolation.** DIGAN can easily interpolate (*i.e.*, fill in interim frames) or extrapolate (*i.e.*, create out-of-frame videos) videos over time by controlling input coordinates. The videos inter- or extra-polated by DIGAN are more natural than those from discrete generative models, as INRs model videos continuously. Table 2 shows that DIGAN outperforms MoCoGAN-HD on all considered inter- and extra-polation scenarios. In particular, DIGAN is remarkably effective for time extrapolation as INRs regularize the videos smoothly follow the scene flows defined by previous dynamics. Figure 4 shows that DIGAN can even extrapolate to 4× longer frames while MoCoGAN-HD fails on the Sky dataset. See Appendix G for more results.

**Non-autoregressive generation.** DIGAN can generate samples of arbitrary time: it enables DIGAN to infer the past or interim frames from future frames or parallelly compute the entire video at once. It is impossible for many prior video generation approaches that sample the next frame conditioned on the previous frames in an autoregressive manner. Figure 5 visualizes the forward and backward prediction results of DIGAN on the TaiChi dataset, conditioned on the initial frame indicated by the yellow box. Here, we find the content and motion latent codes by projecting the initial frames of $t = \{6, 7, 8\}$ and predict the frames of $t \in \{3, \ldots, 11\}$. DIGAN well predicts both past and future motions, *e.g.*, slowly lowering arms. DIGAN can also infer the interim frames from past and future frames, as shown in Appendix E.2. On the other hand, Table 3 shows the generation speed of DIGAN is much faster than its competitors, VideoGPT and MoCoGAN-HD. Different from prior works, we remark that DIGAN can compute multiple frames in parallel; generation can be $N$ times further faster under $N$ number of GPUs. For more analysis about the efficiency, see Appendix I.

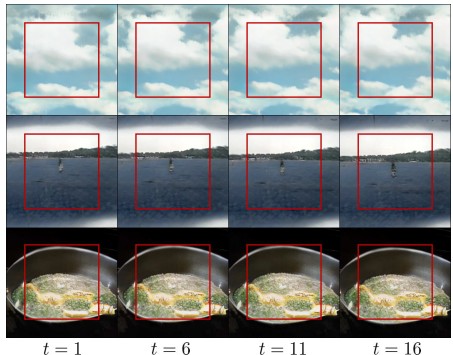

Figure 8: Zoomed-out samples. Red boxes indicate the original frames.

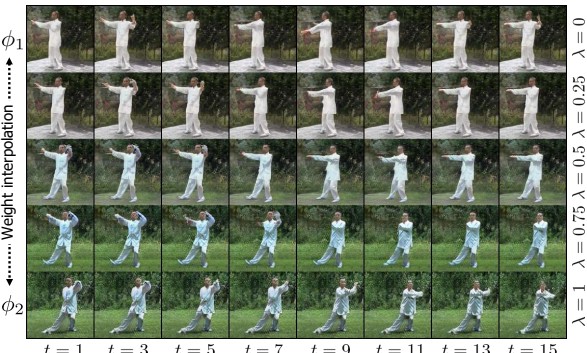

Figure 9: Samples of linearly interpolated INR weights $\phi_i$ over $\lambda$, *i.e.*, $(1 - \lambda)\phi_1 + \lambda\phi_2$ on TaiChi dataset.

Table 5: Ablation study of the generator components: smaller frequency $\sigma_t$, motion vector $z_M$, and non-linearity by MLP $f_M(\cdot)$.

| Freq. $\sigma_t$ | Motion $z_M$ | MLP $f_M(\cdot)$ | FVD ($\downarrow$) |
|---|---|---|---|
| - | - | - | $686\pm25$ |
| ✓ | - | - | $640\pm22$ |
| ✓ | ✓ | - | $585\pm27$ |
| ✓ | ✓ | ✓ | $577\pm21$ |

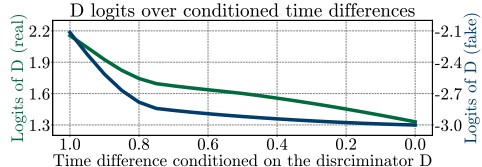

Figure 10: Discriminator logits for a far image pair $(\boldsymbol{i}_0, \boldsymbol{i}_1)$, conditioned over different $\Delta t$.

**Diverse motion sampling.** DIGAN can generate diverse videos from the initial frame by controlling the motion vectors. Figure 6 shows the videos from two random motion vectors on the Sky dataset. Note that the shape of clouds moves differently, but the tree in the left below stays. The freedom of variations conditioned on the initial frame depends on datasets, as discussed in Appendix D.

**Space interpolation and extrapolation.** DIGAN can also inter- and extra-polate videos in the space directions. Figure 7 and Table 4 show the qualitative and quantitative results for space interpolation (*i.e.*, upsampling) on the TaiChi dataset. DIGAN produces $4\times$ higher resolution videos without ad-hoc training tricks, outperforming naïve heuristics such as bicubic interpolation. Figure 8 visualizes the space extrapolation (*i.e.*, zoom-out) results on various datasets. DIGAN creates out-of-box scenes while preserving temporal consistency. See Appendix H for more results.

**INR weight interpolation.** Interpolation of INR parameter sampled from DIGAN produces semantically meaningful videos. Figure 9 visualizes the videos from the linearly interpolated INR parameters on the TaiChi dataset. The videos smoothly vary over interpolation, *e.g.*, cloth color changes from white to blue. It is not obvious as the INR weights lie in a structured, high-dimensional space.

### 4.3 ABLATION STUDIES

We conduct the ablation studies on the components of DIGAN. Table 5 shows that all the proposed generator components: smaller frequency $\sigma_t$, motion vector $z_M$, and non-linear MLP mapping $f_M(\cdot)$ contribute to the generation performance measured by FVD scores on the UCF-101 dataset. We note that the motion vector $z_M$ and MLP $f_M(\cdot)$ remarkably affect FVD when applied solely, but the all-combination result is saturated. On the other hand, Figure 10 verifies that the motion discriminator considers the time difference $\Delta t$ of a given pair of images. Specifically, we provide a far image pair $(\boldsymbol{i}_0, \boldsymbol{i}_1)$, *i.e.*, the first and the last frames, with the time difference $\Delta t$. The motion discriminator thinks the triplet $(\boldsymbol{i}_0, \boldsymbol{i}_1, \Delta t)$ is fake if $\Delta t \approx 0$ and real if $\Delta t \approx 1$, as the real difference is $\Delta t = 1$. Namely, the discriminator leverages $\Delta t$ to identify whether the input triplets are real or fake.

### 5 CONCLUSION

We proposed DIGAN, an implicit neural representation (INR)-based generative adversarial network (GAN) for video generation, incorporating the temporal dynamics of videos. Extensive experiments verified the superiority of DIGAN with multiple intriguing properties. We believe our work would guide various directions in video generation and INR research in the future.

## ETHICS STATEMENT

Video generation has potential threats of creating videos for unethical purposes, *e.g.*, fake propaganda videos of politicians or sexual videos of any individuals. The generated videos, often called DeepFake, have arisen as an important social problem (Westerlund, 2019). To tackle the issue, there have been enormous efforts in detecting fake videos (*e.g.*, Guera & Delp (2018)). Here, we claim that the generation and detection techniques should be developed in parallel, rather than prohibiting the generation research itself. This is because the advance of technology is inevitable, and such prohibition only promotes the technology hide in the dark, making them hard to pick out.

In this respect, the generative adversarial network (GAN) is a neat solution as it naturally trains both generator and discriminator (or detector). Notably, the discriminator trained by GAN can effectively detect the fake samples created by other generative models (Wang et al., 2020). Our proposed video generation method, DIGAN, is also built upon the GAN framework. In particular, we introduced a discriminator that identifies the fake videos without observing long sequences. We believe our proposed discriminator would be a step towards designing an efficient DeepFake detector.

## REPRODUCIBILITY STATEMENT

We describe the implementation details of the model in Appendix A.1, and details of the datasets and evaluation in Appendix A.2. We also provide our code in the supplementary material.

## ACKNOWLEDGMENTS AND DISCLOSURE OF FUNDING

SY thanks Jaeho Lee, Younggyo Seo, Minkyu Kim, Soojung Yang, Seokhyun Moon, Jin-Hwa Kim, and anonymous reviewers for their helpful feedbacks on the early version of the manuscript. SY also acknowledges Ivan Skorokhodov for providing the implementation of INR-GAN. This work was mainly supported by Institute of Information & communications Technology Planning & Evaluation (IITP) grant funded by the Korea government (MSIT) (No.2021-0-02068, Artificial Intelligence Innovation Hub; No.2019-0-00075, Artificial Intelligence Graduate School Program (KAIST)). This work was partly experimented on the NAVER Smart Machine Learning (NSML) platform (Sung et al., 2017; Kim et al., 2018). This work was partly supported by KAIST-NAVER Hypercreative AI Center.

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

## A  IMPLEMENTATION DETAILS

### A.1  MODEL DETAILS

Following the original configuration of INR-GAN (Skorokhodov et al., 2021a), we set the same spatial frequencies $\sigma_x = \sigma_y = \sqrt{10}$ and StyleGAN2 (Karras et al., 2020b) discriminator. We use a small temporal frequency $\sigma_t = 0.25$ to encourage temporal coherence. Here, we remark that the number of frames used in most experiments (16) is much smaller than the image resolution (128), which also affects choosing the $\sigma_t$. In this respect, choosing a larger $\sigma_t$ (but still smaller than $\sigma_x, \sigma_y$) can boost the performance for longer videos, *e.g.*, we use $\sigma_t = 0.5$ for training on 128 frame videos.

In all experiments, we sample $\Delta t := |t_1 - t_2|$ by subtracting the values from two different beta distributions $t_1 \sim \text{Beta}(2, 1), t_2 \sim \text{Beta}(1, 2)$ for both real and generated videos. Such distributions can sample $\Delta t$ diversely (*e.g.*, measured with 10,000 samples, 36.5%, 42.7%, and 20.8% of $\Delta t$ is in the interval of $[0, \frac{1}{3}]$, $[\frac{1}{3}, \frac{2}{3}]$, and $[\frac{2}{3}, 1]$, respectively), and indeed worked well in our experiments.

Like in INR-GAN, we also use a progressive multi-layer perception (MLP) with a factorized multiplicative modulation layers as implicit neural representation architecture. For non-linear mapping $f_M(\cdot)$, we use the 2-layer MLP with leaky ReLU activation and, no bias are applied for motion vectors. We also apply DiffAug (Zhao et al., 2020) to mitigate overfitting from the limited number of videos like in Tian et al. (2021). Specifically, we use all augmentations proposed from DiffAug except CutOut (DeVries & Taylor, 2017), and use the same augmentation to the same video. All other hyperparameters are identical to StyleGAN2 except for $R_1$ regularization coefficient $\gamma$: we use $\gamma = 1$ in all experiments. With these setups, all the experiments are processed with 4 NVIDIA V100 32GB GPUs where it takes at most ~4.4 days to complete.

We note that standard generative adversarial networks (GANs; Goodfellow et al. (2014)) and implicit GANs are in a dual relation as the former samples the input latent while the latter samples the network parameters, which are combined to compute the final outputs. However, the generator of implicit GAN resembles the one of StyleGAN2 in practice since StyleGAN2 injects the input latent to the intermediate layers through the mapping network and modulates the weights with them.

### A.2  DATASET AND EVALUATION DETAILS

**Datasets.** In what follows, we describe datasets that we used for the evaluation of our method. All videos are first pre-processed to video clips of consecutive 16 frames, unless otherwise specified.

- **UCF-101** (Soomro et al., 2012) is a 101-class video action dataset total of 13,320 videos with $320\times240$ resolution. Each video clip is center-cropped to $240\times240$ and resized into $128\times128$ resolution. We conducted two different experiments for a fair comparison: training the model with the train split of 9,357 videos or with all 13,320 videos without the split (following the setup of prior state-of-the-art baselines (Clark et al., 2019; Tian et al., 2021)).

- **Tai-Chi-HD** (Siarohin et al., 2019) is a video dataset total of 280 long videos of people doing Tai-Chi. We use the official link for downloading and cropping the dataset as the rectangle videos of $128 \times 128$ resolution.[5] We use 16 frame video clips of stride 4 (*i.e.*, skip 3 frames after the chosen frame) for dynamic motion. We use all data without the split on training. We exclude the following videos due to the broken link: 8xSkbMUpegs, RCiy2FYViEg, iFMbu9-Mejc, ceoe2fz648U, 6jHyn4z0KLk, VMSqvTE90hk, xmwGBXYofEE, Dn0mNZmAh2k, VhprHat04dk, KYdyIdusD0g, EaEZVfhn07o, ⌐L745tFFmCQ, ytT4iU7h-A8, 5ujMzSyHO⌐8, JdiIQg47Wc4, aAwbJ9MO91I, and ⌐XRyc2kiTlM.

- **Sky Time-lapse** (Xiong et al., 2018) is a collection of sky time-lapse total of 5,000 videos. We use the same data pre-processing following the official link.[6] We use the train split for training the model and test split for the evaluation, following the setups in prior works.

- **Kinetics-600** (Carreira et al., 2018) is a large-scale 600-class video action dataset, consists of a total of 495,547 videos. We sub-sampled a food subclass in the dataset to train the model, where we follow the list of such a subclass from Weissenborn et al. (2020), namely: (baking,

---

[5] https://github.com/AliaksandrSiarohin/first-order-model
[6] https://github.com/weixiong-ur/mdgan

`barbequing`, `breading`, `cooking`, `cutting`, `pancake`, `vegetables`, `meat`, `cake`, `sandwich`, `pizza`, `sushi`, `tea`, `peeling`, `fruit`, `eggs`, and `salad`. We use train split for the model training and use the validation set for the evaluation. Note that we only use these classes, different from Weissenborn et al. (2020) to train the model with the whole dataset.

**Evaluation metrics.** We follow the prior setups for evaluation for a fair comparison. We use the C3D network (Tran et al., 2015) pre-trained on Sports-1M (Karpathy et al., 2014) and fine-tuned on UCF-101 datasets for the Inception score (IS; Salimans et al. (2016)). We use the official TGAN (Saito et al., 2017) implementation for computing IS: the score is evaluated over 10,000 generated videos.[7] We note this metric does not use the Inception network (Szegedy et al., 2016) have been used in evaluating IS in image generation, but only the formula for evaluation is identical.

For Fréchet video distance (FVD; Unterthiner et al. (2018)), and kernel video distance (KVD; Unterthiner et al. (2018)), we use the I3D network trained on Kinetics-400 (Kay et al., 2017). All evaluations are done by averaging 10 runs of scores computed from 2,048 sampled real and generated videos, following the setup in TGANv2 (Saito et al., 2020).

**Forward and backward prediction.** For the forward and backward prediction in Figure 5, one should project the given frame into the latent code. To this end, we follow StyleGAN2 projection procedure Karras et al. (2020a) and optimize for 20,000 iteration.

**Generation time.** To fairly compare the generation time with the baselines in Table 3, we utilize the same machine and stop other processes. We used Intel(R) Xeon(R) CPU E5-2630 v4 @ 2.20GHz and a Titan XP GPU for the measurement.

**Space extrapolation.** We generate the video by 2D grid of range $[-0.25, 1.25]^2$.

## B   BASELINES

In this section, we explain video generation baselines we used for evaluating DIGAN at a high level.

- **VGAN** (Vondrick et al., 2016) extends image generative adversarial network (GAN; (Goodfellow et al., 2014)) by replacing 2D spatial architecture with 3D spatio-temporal architecture.
- **TGAN** (Saito et al., 2017) separates the spatial (image) generator and temporal (latent dynamics) build on Wasserstein GAN (Arjovsky et al., 2017) model for image generation.
- **MoCoGAN** (Tulyakov et al., 2018) proposes GAN to disentangle videos by the content of single latent and motion of stochastic latent trajectory and generate these two components.
- **ProgressiveVGAN** (Acharya et al., 2018) progressively generates videos both in spatial and temporal directions, like pregressive image generation in ProgressiveGAN (Karras et al., 2017).
- **LDVD-GAN** (Kahembwe & Ramamoorthy, 2020) proposes a efficient video discriminator for training video GANs by considering its kernel dimension to be low.
- **VideoGPT** (Yan et al., 2021) compresses videos as a sequence of discrete latent vectors via vector-quantized variational auto-encoder (van den Oord et al., 2017), then trains autoregressive Transformer model (Vaswani et al., 2017) with these latent sequences.
- **TGANv2** (Saito et al., 2020) designs a new video GAN of computationally efficient generator and discriminator by proposing sub-modules on each component.
- **DVD-GAN** (Clark et al., 2019) uses spatial and temporal discriminators, where the input of the temporal discriminator is spatially down-sampled to reduce the computational bottleneck.
- **MoCoGAN-HD** (Tian et al., 2021) leverages a pre-trained image generator to additionally train a motion generator on the image latent space to synthesize the video with those two generators.

---

[7]https://github.com/pfnet-research/tgan

## C  ADDITIONAL RELATED WORK

**Video prediction.** A related but distinct area, video prediction, aims to forecast future video frames from early frames (Srivastava et al., 2015; Finn et al., 2016; Denton & Birodkar, 2017; Babaeizadeh et al., 2018; Denton & Fergus, 2018; Lee et al., 2018; Villegas et al., 2019; Kumar et al., 2020; Franceschi et al., 2020; Luc et al., 2020; Lee et al., 2021). By restricting the distribution conditioned on initial frames, video prediction models often show better visual quality than video generation (Babaeizadeh et al., 2021). Also, video prediction models employ image-to-image architectures as they only need to modify the initial frames regarding the motion, while video generation models use latent-to-image architectures. While our primary focus is video generation, one can utilize our model for video prediction by projecting initial frames into the corresponding latent codes.

**Video synthesis with motion and content decomposition.** For better video generation or prediction with controllability, several works have proposed methods to decompose the motion and contents. Specifically, they model videos with a content vector and a sequence of motion vectors from different subspaces, incorporating distinct motion and content encoders for a prediction (Villegas et al., 2017; Hsieh et al., 2018) or generators for a generation (Tulyakov et al., 2018; Tian et al., 2021; Munoz et al., 2021). Similarly, we utilize two generators, but the motion is represented as a single vector of the weight of implicit neural representations rather than a sequence of latents.

**Concurrent work.** A concurrent work, StyleGAN-V (Skorokhodov et al., 2021b), also focuses on generative modeling of videos by interpreting videos as continuous signals. Moreover, it shares many similar ideas with DIGAN, *e.g.*, a motion discriminator without 3D convolution networks. However, it has a core difference to DIGAN; DIGAN treats videos as spatiotemporally continuous signals, while StyleGAN-V extends StyleGAN2 (Karras et al., 2020b) by interpreting videos as temporally continuous signals (but not spatially continuous), *i.e.*, video frames are discrete 2D grids.

# D    LATENT DYNAMICS OF MOTION FEATURES

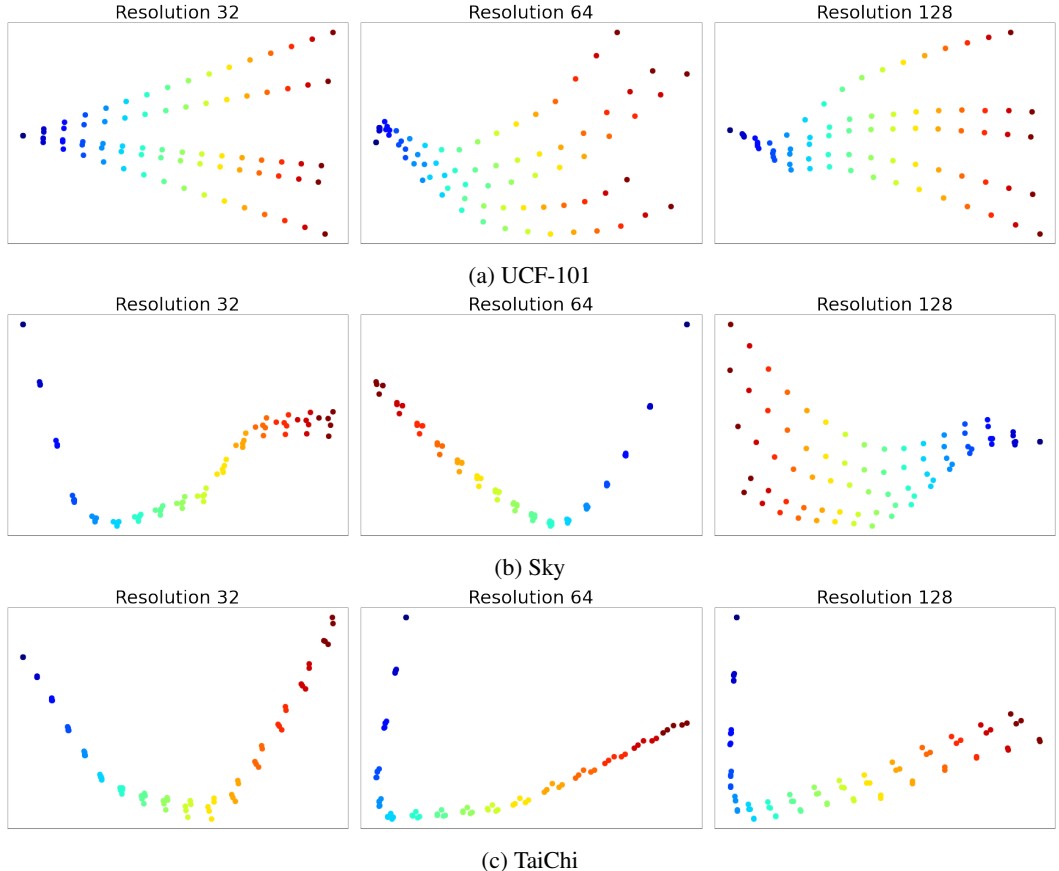

Figure 11: Latent trajectories of the motion features on UCF-101, Sky, and TaiChi datasets. Resolution denotes the location of motion features injected into the progressive generator; lower-resolution controls the high-level semantics, and higher-resolution controls the low-level variations. Dot colors gradually changes from blue ($t = 0$) to red ($t = 1$), and 5 random motion vectors are sampled. The features are projected onto 2D space via principal component analysis (PCA) for visualization.

Figure 11 visualizes the latent dynamics of motion features on various datasets. Note that the freedom of motion features vary for different datasets: UCF-101 allows high-level (resolution 32) and low-level (resolution 128) variations, Sky only allows low-level variations, and TaiChi does not permit the variations much. Intuitively, the next move of TaiChi is mostly determined by the prior frames: the movement follows the pre-defined gestures. Sky permits some variations as shown in Figure 6, but the wind direction determines the global motion. UCF-101 has the largest freedom of motion as the dataset contains videos of various actions and diverse objects.

# E   MORE EXAMPLES ON VIDEO PREDICTION

## E.1   MORE BACKWARD PREDICTION

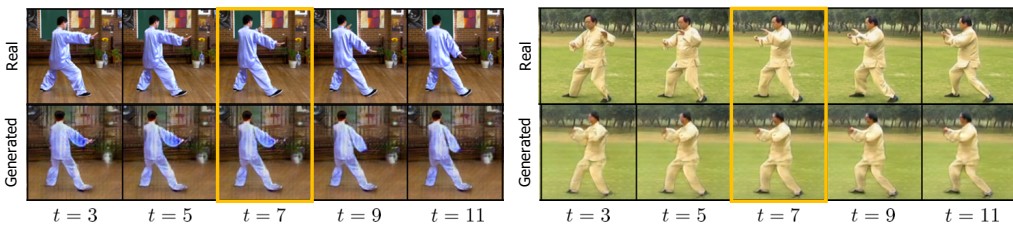

Figure 12: More examples on forward and backward prediction results of DIGAN. Yellow box indicates the given frame.

## E.2   INTERMEDIATE SCENE PREDICTION

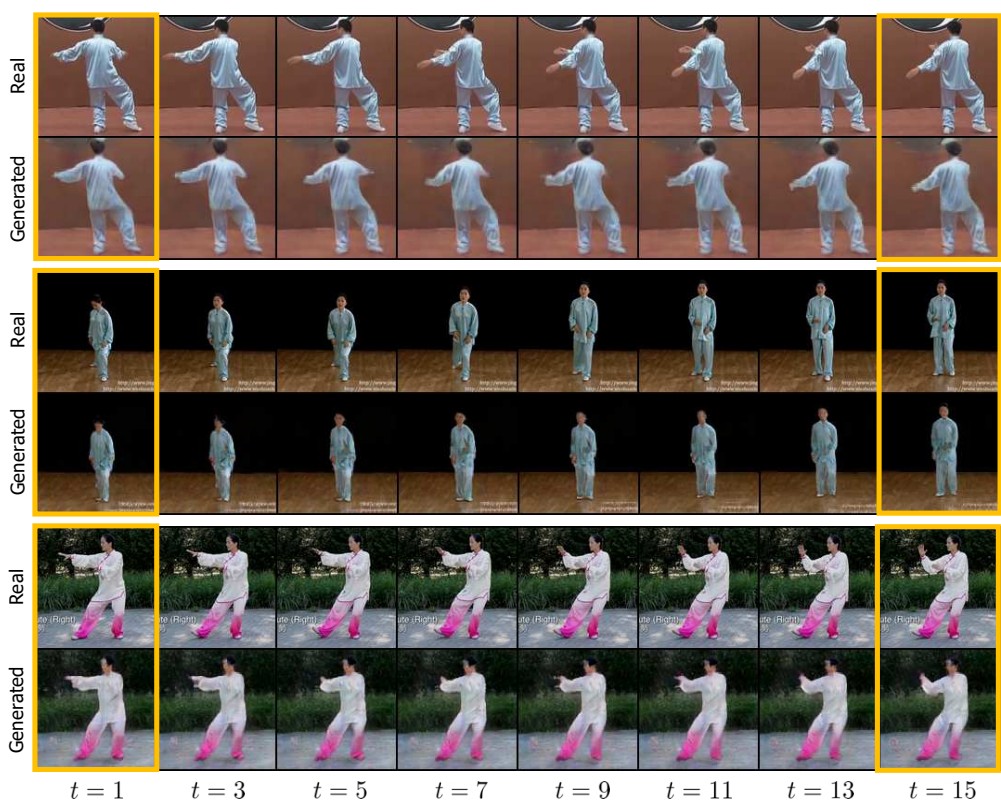

Figure 13: Intermediate scene (*i.e.*, frame) prediction results of our method, DIGAN. The yellow boxes indicate the given initial and last frames.

We show that our proposed method, DIGAN, can predict the intermediate frames between the two frames of the different time steps. To this end, we find the inverse content and motion latent codes by projecting two given frames into the generator (we follow the StyleGAN2 projection procedure (Karras et al., 2020b) and optimize 20,000 iteration for the projection). As shown in Figure 13, DIGAN can well predict the intermediate dynamics even the given frames are somewhat far apart. This result implies that DIGAN indeed has learned the dynamics of the ground truth distribution.

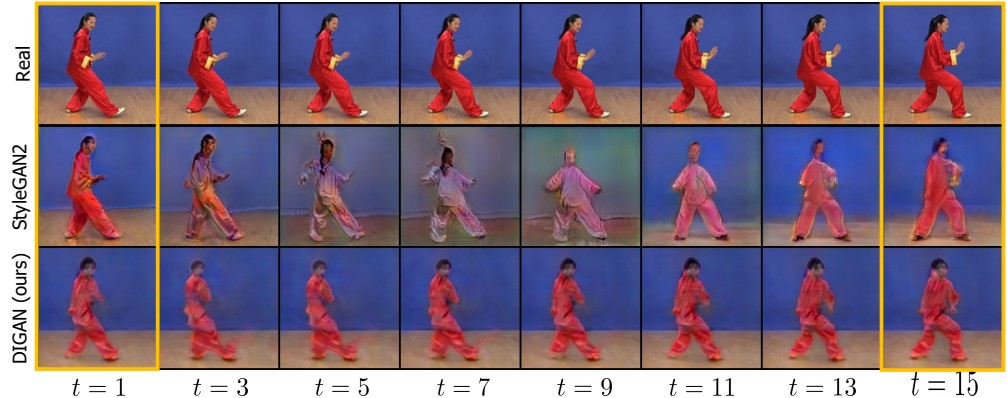

Figure 14: Comparison of intermediate scene (*i.e.*, frame) prediction results by StyleGAN2 and our method, DIGAN. The yellow boxes indicate the given initial and last frames.

Table 6: SSIM of intermediate scenes predicted by StyleGAN2 and our method, DIGAN.

|  | SSIM ($\uparrow$) |
| --- | --- |
| StyleGAN2 latent interpolation | 0.5639 |
| DIGAN (ours) | 0.6753 |

We also report structural similarity index measure (SSIM) to quantitatively compare predicted videos from DIGAN and linearly interpolated image sequences in StyleGAN2 latent space. For StyleGAN2, we use the official StyleGAN2 inversion method, namely, we project each of the initial and final frames to the StyleGAN2 latent space trained on the Taichi dataset. After that, we linearly interpolate on these two projected latent vectors. As shown in Table 6, DIGAN shows better prediction than StyleGAN2 interpolation. Moreover, as shown in Figure 14, one can observe that DIGAN preserves semantics like background across the temporal direction, while the StyleGAN2 latent interpolation cannot.

## F MORE EXAMPLES FOR LONG VIDEO GENERATION

→ Follow the arrow direction, and move to the next line at the end

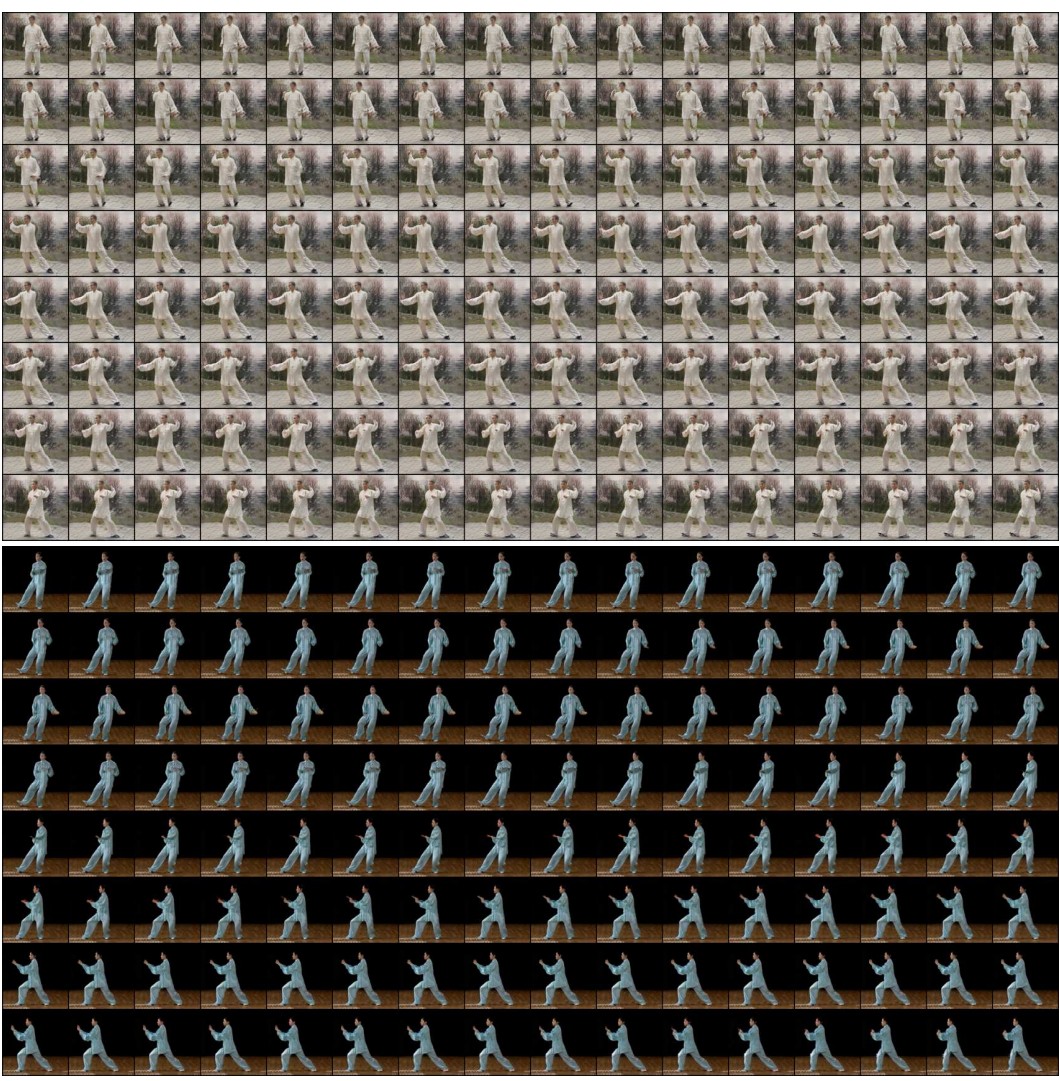

Figure 15: Additional 128 frame videos of 128×128 resolution by DIGAN, on the TaiChi dataset.

## G  MORE EXAMPLES FOR TIME EXTRAPOLATION

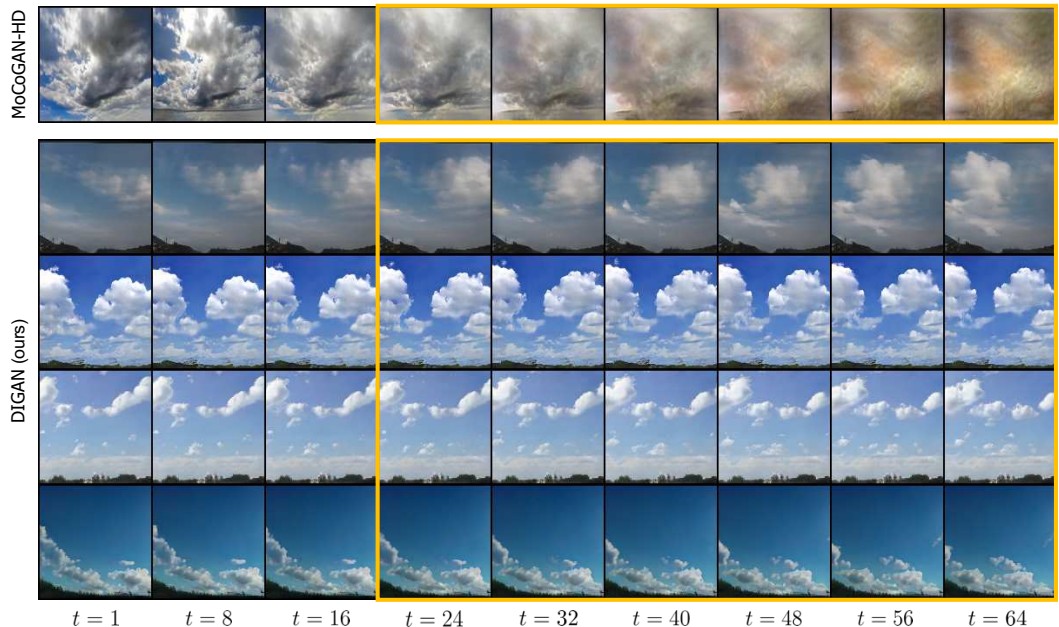

Figure 16: Additional time extrapolation videos of MoCoGAN-HD and DIGAN, trained on 16 frame videos of 128×128 resolution on the Sky dataset. Yellow box indicates the extrapolated frames.

## H  MORE EXAMPLES FOR SPACE EXTRAPOLATION

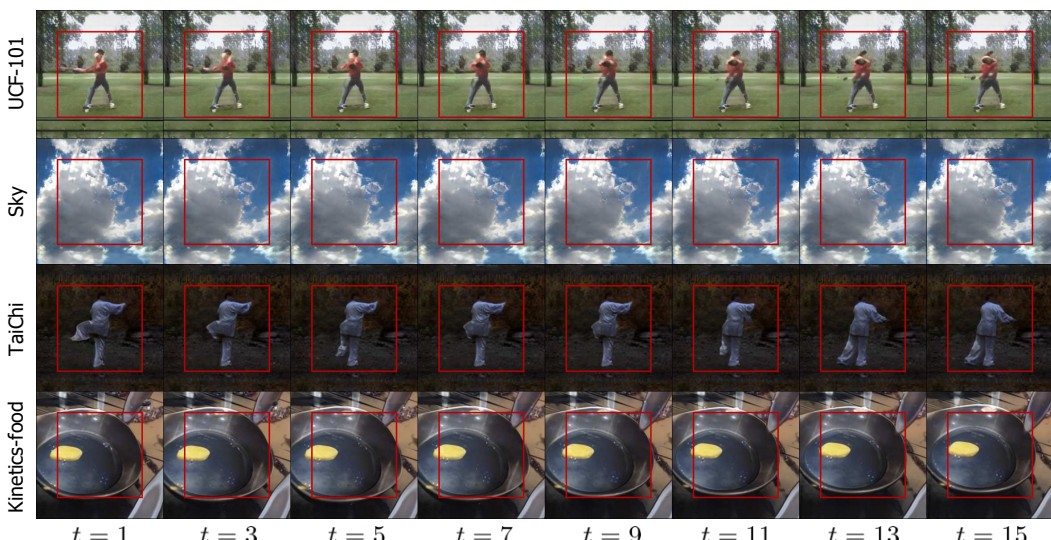

Figure 17: Additional zoomed-out samples. Red boxes indicate the original frames.

## I  EFFICIENCY OF DIGAN

In addition to the computational efficiency on inference (in Table 3), we provide the analysis of our method on efficiencies of diverse aspects, including computation, energy, memory, and time efficiency. Here, we mainly compare our method with MoCoGAN-HD (Tian et al., 2021), which is an energy-efficient (*i.e.*, shorter GPU days for training) and state-of-the-art method on the generation quality. All experiments are performed under the same machine (NVIDIA V100 32GB GPUs).

- **Computational efficiency.** We report floating point operations per second (FLOPS)[8] of video GAN generators (following Liu et al. (2021)). The DIGAN generator requires 147.9 GFLOPS to generate a 16 frame video of 128×128 resolution, 4.6 times smaller than 682.3 GFLOPS of the MoCoGAN-HD generator.
- **Energy efficiency on training (GPU days for training).** The FVD value of DIGAN trained on UCF-101 for 8 GPU days achieves 689±24, which is better than 838 of MoCoGAN-HD trained for 16 GPU days. This implies that DIGAN is (at least 2 times) more energy-efficient than MoCoGAN-HD.
- **Memory efficiency on training.** We report the memory size per GPU to load 16 frame videos of 128×128 resolution of batch size 4 for training video GANs. DIGAN allocates 9.7GB memory, 2.9 times smaller than 28.0GB of MoCoGAN-HD.
- **Time efficiency on inference.** Per each GPU, DIGAN discriminators can (forward) compute 195.6 video clips/sec, 3.2 times larger than 60.3 video clips/sec of MoCoGAN-HD discriminators with 128 frame videos of 128×128 resolution and a batch size of 16. It confirms that both the generator and discriminator of our method are much more time-efficient; recall that we already demonstrated that our generator is ∼2.3 times faster per each GPU in Table 3.

In summary, DIGAN is more computation-efficient ($> 4.6$ times), energy-efficient on training ($> 2$ times), memory-efficient on training ($> 2.9$ times), and time-efficient on inference ($> 3.2$ times) under each machine setup above. Such gaps stem from the fact that our method utilizes: (a) INR-based generator without time-consuming autoregressive modeling and (b) 2D convolutional discriminator rather than computationally heavy 3D convolutional discriminator. Finally, we emphasize that DIGAN can synthesize multiple frames of a video in parallel (which is impossible under prior autoregressive or convolutional network-based methods), resulting in $N$ times improved time-efficiency to generate a single video under $N$ number of GPUs. It can be particularly important for synthesizing extremely long videos; since it requires tremendous time if utilizing prior methods that are impossible to compute frames in parallel. Like this, the superiority of computational efficiency of DIGAN can be further dramatically improved in the presence of multiple GPUs.

## J  EFFECT OF CONTENT VECTOR FOR GENERATING MOTION

Table 7: Effect of the motion vector $z_I$ for generating the motion. We report the mean and standard deviation of the FVD values over 10 runs.

|  | UCF-101 | TaiChi |
|---|---|---|
| Motion only | 596±26 | 147.9±4.9 |
| Motion + content (ours) | 577±21 | 128.1±4.9 |

We provide the result when only the motion vector is used as input (*i.e.*, without the content vector); we report the FVD values of the models on the UCF-101 and the TaiChi dataset. The result in Table 7 confirms that the consideration of content vectors improves the result, as video motions often depend on the content (*i.e.*, initial frame determines the possible future motions).

---

[8] https://github.com/facebookresearch/fvcore

# K  CLASS-CONDITIONAL GENERATION OF DIGAN

Table 8: IS, FVD, and KVD values of video generation models on the UCF-101 dataset. ↑ and ↓ imply higher and lower values are better, respectively. Subscripts denote standard deviations, and bolds indicate the best results.

| Method | IS ($\uparrow$) | FVD ($\downarrow$) | KVD ($\downarrow$) |
|---|---|---|---|
| DVD-GAN | $32.97_{\pm 1.7}$ | - | - |
| TSB | $42.79_{\pm 0.63}$ | - | - |
| DIGAN (ours) | $\mathbf{59.68_{\pm 0.45}}$ | $465_{\pm 12}$ | $39.6_{\pm 2.9}$ |

We provide both qualitative and quantitative results of DIGAN on a class-conditional setup to compare several recent works on video synthesis that reported class-conditional generation results, such as DVD-GAN (Clark et al., 2019) and TSB (Munoz et al., 2021). Specifically, we report class-conditional generation results trained on UCF-101 in Table 8 and the project page,[9] which demonstrate the superiority of DIGAN even on the conditional setup compared to existing baselines.

---

[9]https://sihyun-yu.github.io/digan/

