# OpenReview forum: "Generating Videos with Dynamics-aware Implicit Generative Adversarial Networks"
_ICLR.cc/2022/Conference — ICLR 2022 Poster_

### Official Review · Reviewer_vhpn · 2021-10-31

**Correctness:** 4
**Technical Novelty And Significance:** 3
**Empirical Novelty And Significance:** 3
**Recommendation:** 10
**Confidence:** 5

**Main Review:**

Strengths:

1. This work proposes a simple extension of INRs to video generation.
INRs have thus far remained in the realm of image generation. To my knowledge, this is the first work to extend this approach to video.
In hindsight, this may seem like a trivial extension, but the introduction of INRs to video generation provides a solution to many hard problems in video generation work.

2. As a result of being the first work to present INRs as a viable method for tackling video generation, it benefits from the efficiency advantages inherent to using INRs for videos, which include but are not limited to: long horizon video generation (can predict up 128 frames in future at a resolution of 128x128), time interpolation and extrapolation (can predict interim frames and also predict far out into the future), non-autoregressive generation (can generate frames for specific time points of interest without generating intermediate frames), space interpolation and extrapolation (can up sample and zoom out of generated video)

3. The work is well motivated, written and benchmarks across all the standard video generation datasets. It also includes the relevant ablation studies. It is also timely, given the need for more efficient video GAN models.



Weaknesses:
1. This work suffers from one fatal flaw. Some of its experiments train the proposed model on the full dataset, not the training split as in prior work, yet it presents results comparing performance to models trained on only the training split of the dataset. I have to admit that I missed this as a reviewer of the prior work that was submitted and accepted to ICLR last year, but since it was not explicitly stated, it was assumed that the prior work followed proper experimentation procedure when benchmarking against prior art. Particularly for UCF101, the results where models train on the full 13320 videos should be explicitly differentiated from models that train on the "trainlist01" training split of UCF101 comprised of only 9537 videos. To allow for honest and accurate comparison to the majority of prior work, I would strongly encourage the authors to benchmark models trained on the training data split for the related datasets, in particular, the "trainlist01" split of the UCF101 dataset. They should also highlight and mark (e.g. star) results that come from models trained on a full dataset, that compare their results to prior art that only trains on a split of that dataset.

2. This work is lacking when it comes to experiments comparing the compute, memory, time or energy efficiency of the proposed approach. This would greatly strengthen the proposed contribution since it is so naturally amenable to great gains in efficiency across all four metrics.

**Summary Of The Paper:**

The paper proposes a video generation approach based on implicit neural representations (INRs). They construct a "dynamics-aware" GAN; where by the generator model used derives from the coordinate-based models used to capture continuous representations of images in prior  INR work. The discriminator architecture derives from the 2D discriminator traditionally used in image GAN research; they simply expand the input channels from the traditional 3 channels (i.e. rgb) used; to 7 input channels. This discriminator processes a stack of images from one video simultaneously, two frames from different time points in the video and their difference image.

**Summary Of The Review:**

This is good work, a simple idea extended from the image to the video domain. It suffers a fatal flaw that prevents me from recommending it for acceptance.

In its current form, this paper is "marginally below the acceptance threshold". If the issues with benchmarking are resolved, then I will recommend it for acceptance. If additional experiments exploring the efficiency of the proposed approach are provided, then I will strongly recommend it for acceptance.


------

***EDIT****

The authors have sufficiently addressed all of my concerns (and those of the other reviewers it seems).
I strongly recommend this paper for acceptance to ICLR22, it is a meaningful step in a promising direction for long-horizon GAN-based video generation.

---

> ### Author Response · Authors · 2021-11-12
> **Response to Reviewer vhpn (2/2)**
>
> **[C2] More experimental results on efficiency of the method.**
>
> Thank you for your constructive suggestion. In addition to the computational efficiency on inference (in Table 3), we added the analysis of our method on efficiencies of diverse aspects, including computation, energy, memory, and time efficiency (we add Appendix I in the revision). Here, we mainly compared our method with MoCoGAN-HD [1], which is an energy-efficient (i.e., shorter GPU days for training) and state-of-the-art method on the generation quality. All experiments are performed under the same machine (NVIDIA V100 32GB GPUs).
>
> - **Computational efficiency.** We report floating point operations per second (FLOPS) of video GAN generators (following [4]). The DIGAN generator requires 147.9 GFLOPS to generate a 16 frame video of 128$\times$128 resolution, 4.6 times smaller than 682.3 GFLOPS of the MoCoGAN-HD generator.
> - **Energy efficiency on training (GPU days for training).** The FVD value of DIGAN trained on UCF-101 for 8 GPU days achieves 689$\pm$24, which is better than 838 of MoCoGAN-HD trained for 16 GPU days. This implies that DIGAN is (at least 2 times) more energy-efficient than MoCoGAN-HD.
> - **Memory efficiency on training.** We report the memory size per GPU to load 16 frame videos of 128$\times$128 resolution of batch size 4 for training video GANs. DIGAN allocates 9.7GB memory, 2.9 times smaller than 28.0GB of MoCoGAN-HD.
> - **Time efficiency on inference.** Per each GPU, DIGAN discriminators can (forward) compute 195.6 video clips/sec, 3.2 times larger than 60.3 video clips/sec of MoCoGAN-HD discriminators with 128 frame videos of 128$\times$128 resolution and a batch size of 16. It confirms that both the generator and discriminator of our method are much more time-efficient; recall that we already demonstrated that our generator is $\sim$2.3 times faster per each GPU in Table 3.
>
> In summary, DIGAN is more computation-efficient ($>$ 4.6 times), energy-efficient on training  ($>$ 2 times), memory-efficient on training ($>$ 2.9 times), and time-efficient on inference ($>$ 3.2 times) under each machine setup above. Such gaps stem from the fact that our method utilizes: (a) INR-based generator without time-consuming autoregressive modeling and (b) 2D convolutional discriminator rather than computationally heavy 3D convolutional discriminator. Finally, we emphasize that DIGAN can synthesize multiple frames of a video in parallel (which is impossible under prior autoregressive or convolutional network-based methods), resulting in $N$ times improved time-efficiency to generate a single video under $N$ number of GPUs. It can be particularly important for synthesizing extremely long videos; since it requires tremendous time if utilizing prior methods that are impossible to compute frames in parallel. Like this, the superiority of computational efficiency of DIGAN can be further dramatically improved in the presence of multiple GPUs.
>
> ---
>
> [1] Tian et al., A Good Image Generator is What you Need for High Resolution Video Synthesis, ICLR 2021\
> [2] Clark et al., Adversarial Video Generation on Complex Datasets, 2019\
> [3] Saito et al., Train Sparsely, Generate Densely: Memory-Efficient Unsupervised Training of High-Resolution Temporal GAN, IJCV 2020\
> [4] Liu et al., Content-Aware GAN Compression, CVPR 2021

---

> ### Author Response · Authors · 2021-11-12
> **Response to Reviewer vhpn (1/2)**
>
> We sincerely appreciate your efforts and insightful comments to improve the manuscript. We respond to each of your comments one-by-one in what follows. In the revised draft, we mark our major revisions by “blue”. We also provide additional examples of videos in our project page: https://iclr2022-digan.github.io.
>
> ---
>
> **[C1] Experimental results with only train split on UCF-101 for a fair comparison.**
>
> Thank you for pointing out this important issue! As you mentioned, we used the entire dataset following the recent experimental setup in [1, 2] for fair comparison with their results. But, we totally agree with your concern that this might not be fair with other prior works. To address the concern, we report the performance of our model under both split setups; see the below table (we also update Table 1 and Appendix A.2 in the revision). Namely, we now additionally provide the result on UCF-101 done only with the “train split” (following the setup in [3]). The result confirms that our method achieves the state-of-the-art scores in both cases. Moreover, we also explicitly stated how the baseline and our results are achieved, i.e., whether the model is trained with the entire dataset or only with the train split. We think our results would be a useful guideline when other researchers pursue a similar task in the future.
>
> \begin{array}{l cc}
> \hline
>   & \text{IS }(\uparrow) & \text{FVD }(\downarrow) \newline
> \hline
> \textit{Train split} & & \newline
> \hline
> \text{VGAN} & \phantom{0}8.31\small{\pm.09} & - \newline
> \text{TGAN} & 11.85\small{\pm.07} & - \newline
> \text{MoCoGAN} & 12.42\small{\pm.07} & - \newline
> \text{ProgressiveVGAN} & 14.56\small{\pm.05} & - \newline
> \text{LDVD-GAN} & 22.91\small{\pm.19} & - \newline
> \text{VideoGPT} & 24.69\small{\pm.30} & - \newline
> \text{TGANv2} & 28.87\small{\pm.67} & {1209} \small{\pm28} \newline
> \text{DIGAN (ours)}  & \mathbf{29.71\small{\pm.23}} & \phantom{0}\mathbf{{655} \small{\pm22}} \newline
> \hline
> \textit{Train}+\textit{test split} & & \newline
> \hline
> \text{DVD-GAN} & 27.38\small{\pm.53} & - \newline
> \text{MoCoGAN-HD} & 32.36\small{\phantom{0}\phantom{0}\phantom{0}\phantom{0}} & 838 \small{\phantom{0}\phantom{0}\phantom{0}\phantom{0}} \newline
> \text{DIGAN (ours)} & \mathbf{32.70\small{\pm.35}} & \mathbf{{577} \small{\pm21}} \newline
> \hline
> \end{array}

---

> ### Author Response · Authors · 2021-11-22
> **A gentle reminder**
>
> Dear Reviewer vhpn,
>
> Thank you for your time and efforts in reviewing our paper.
>
> We kindly remind that the discussion period will end in 30 hours or so.
>
> We sincerely hope that our response and results of the supporting experiments successfully clarify your concerns.
>
> We just wonder whether we could have the last chance to address your further concerns or questions (if you have any).
>
> Thank you very much!
>
> Authors

---

> ### Author Response · Authors · 2021-11-23
> **Thank you for the update and careful reading on our rebuttal**
>
> We are happy to hear that our response could help to address your concerns !
>
> We appreciate not only your strong support on our paper, but also your careful reading on our draft and rebuttal (even those for other reviewers).
>
> Due to your valuable and constructive suggestions (based on your expertise), we do believe that our paper is much improved.
>
> If you have any further questions or concerns, please do not hesitate to let us know.
>
> Thank you very much,
>
> Authors

---

### Official Review · Reviewer_Tj68 · 2021-11-01

**Correctness:** 4
**Technical Novelty And Significance:** 3
**Empirical Novelty And Significance:** 3
**Recommendation:** 8
**Confidence:** 4

**Main Review:**

## General Thoughts
The paper is interesting and the method propose definitely shows promise. Applying INRs to this task seems like an excellent direction, and avoids many computational problems of video generation. I would have loved to see a more compelling set of visual examples to highlight the possibilities of the technique, and a potential comparison to other works that have trained on larger portions of Kinetics-600.

## Strengths
- The proposed alternative has several positive poperties, such as non-autoregessive generation.
- Decomposition of spatial and temporal factors in the INR
- Good variety in datasets tested
- Strong improvements in quantitative metrics
- INRs seem like a very good approach for this task, love the concept

## Weaknesses
- A few typos and writing could be improved (last paragraph of pages 4 and 5 for example)
- Would love to see an ablation study showing that including the motion + content vector as input to the motion generator (instead of just the motion vector) improves performance
- The idea of comparing two frames and their time difference is nice, but its power is only truly leveraged if many different $\Delta t$ are shown to the discriminator - is this the case? How are you handling this during training?
- Would love to see more results of the effects of sampling different motion vectors - Figure 6 is not compelling enough to believe the claim that the presented motion vector setup is beneficial. There is only one example in the anonymized webpage.
- Why was only TaiChi used for the interpolation task? This technique should handle interpolation much better than its counterparts, and showing more results there seems like a winning move. Why were the other datasets omitted?
- Fig 5 is not very compelling, would modify or add more examples. Would have loved to see more examples in the appendix or webpage.
- Comparisons to MoCoGAN-HD's 1024x1024 videos would have been nice, to really test the extrapolation abilities of INRs


## Questions
- What is Lambda in figure 9? Can't find it in the main text
- Figure 10: compelling, but I'm left to wonder if the discriminator does this even with a fake pair of frames: For a fake video, when the frames are far apart, we'd like the discriminator to still say that the input is fake, right?
- Discriminator input channels increased from 3 to 7 -> why do this instead of siamese nets to compare the two frames?




**Summary Of The Paper:**

This paper leverages the implicit neural representations paradigm to build generative adversarial networks for video generation. Implicit neural representations encode continuous signals into parametrized neural networks, and the authors claim that this mitigates the issue related to the inefficient modeling of videos as 3d tensors of RGB values. Their GAN includes an INR-based video generator, and a motion discriminator that is more efficient at identifying unnatural motions. Their approach yields FVD improvements in various existing datasets.

**Summary Of The Review:**

UPDATE: The provided rebuttal answers most of my concerns, and my confidence about the strength of the paper has increased. I am updating my recommendation to 8 - accept.

The paper discusses a very relevant concept in video generation: utilizing INRs to improve performance and efficiency. The authors show strong quantitative results and a few architectural novelties. The work is solid, but there is a worrying lack of compelling qualitative examples to back up certain claims. If the authors can provide more examples, I believe this could be a strong addition to ICLR.

---

> ### Author Response · Authors · 2021-11-16
> **Response to Reviewer Tj68 (2/2)**
>
> **[C6] Better illustrations of video prediction (Figure 5).**
>
> Thank you for the suggestion! To incorporate your comment, we modify Figure 5 with another example and provide more examples of video prediction in Appendix E.
>
> ---
>
> **[C7] Comparison with high-resolution videos in MoCoGAN-HD.**
>
> We note that high-resolution videos from MoCoGAN-HD are “cross-domain synthesis” results, i.e., video synthesis with the target domain image generator (trained with “high-resolution image” dataset) using the motion generator (trained with video datasets not related to high-resolution image dataset). Since DIGAN directly trains video GANs from the given video datasets, a direct comparison of DIGAN and provided results from MoCoGAN-HD is impossible. A fair comparison would be made by training our method and MoCoGAN-HD using only high-resolution video datasets from scratch. However, to our best knowledge, there are no well-curated high-resolution video datasets like FFHQ in the image domain, which makes the comparison difficult. Nevertheless, since DIGAN can spatially interpolate, we provide 1024$\times$1024 resolution videos spatially interpolated from 128$\times$128 videos on the project page; they are temporally coherent even at 8$\times$ spatial interpolation.
>
> ---
>
> **[Q1] Missing description of $\lambda$ in Figure 9.**
>
> Thank you for pointing this out. $\lambda$ denotes the linear interpolation coefficients for two different INR weights $\phi_1$ and $\phi_2$, i.e., $(1 - \lambda)\phi_1 + \lambda \phi_2$. We clarify this by updating Figure 9 to indicate $\phi_1$ and $\phi_2$ explicitly and fix its caption as “Samples of linearly interpolated INR weights $\phi_i$ over $\lambda$, i.e., $(1-\lambda)\phi_1 + \lambda\phi_2$ on TaiChi dataset”  in the revision.
>
> ---
>
> **[Q2] Ablation study of $\Delta t$ on the discriminator with fake videos.**
>
> The discriminator shows a similar trend to Figure 10 for the generated images: we remind you that Figure 10 shows that the discriminator considers the time difference $\Delta t$ for a given pair of real images for distinguishing real or fake. Specifically, we report the discriminator logit values of the pair of fake images with decreasing $\Delta t$ where the ground-truth time difference is 1 (i.e., the first and the last frames). The logit value of this fake image pair decreases if $\Delta t$ decreases, e.g., the logit value is -2.12 if $\Delta t = 1$ while the value becomes -3.00 if $\Delta t = 0$. It implies that the time difference correctly affects the discriminator like the pair of real images. Note that these logit values are overall smaller than those from the pair of real images since the realness of frames may also affect the discriminator. We update Figure 10 to include this result in the revision.
>
> ---
>
> **[Q3] Reason for the design choice of the current video discriminator.**
>
> We design our discriminator by increasing the input channel since it is a straightforward approach and does not significantly increase both the computation and the memory. Nonetheless, we agree that other forms of discriminator can be designed, such as utilizing siamese nets, as you suggested. Developing different forms of discriminator should be an interesting future direction.

---

> ### Author Response · Authors · 2021-11-16
> **Response to Reviewer Tj68 (1/2)**
>
> We sincerely appreciate your efforts and insightful comments to improve the manuscript. We respond to each of your comments one-by-one in what follows. In the revised draft, we mark our major revisions by “blue”. We also provide additional examples of videos in our project page: https://iclr2022-digan.github.io.
>
> ---
>
> **[C1] Editorial comment.**
>
> Thank you for your suggestion. We fix typos (e.g., “shows”$\rightarrow$“show” in page 9) and improve the writing (e.g., modify the last paragraph of page 4 and 5) in the revision. If you could guide us on which aspects of writing are particularly to be further improved, it would be very much appreciated!
>
> ---
>
> **[C2] Ablation study of the content vector to the motion generator.**
>
> To answer your question, we provide the result when only the motion vector is used as input (i.e., without the content vector); we report the FVD values (lower is better) of the models on the UCF-101 and the TaiChi dataset. The result confirms that the consideration of content vectors improves the result, as video motions often depend on the content (i.e., initial frame determines the possible future motions). We include the results in Appendix J.
>
> \begin{array}{lcc}
> \hline
>  & \text{UCF-101} & \text{TaiChi} \newline
> \hline
> \text{Motion-only} & 596 \small{\pm 26} & 147.9 \small{\pm 4.9}   \newline
> \text{Motion + content (ours)} & 577 \small{\pm 21} & 128.1 \small{\pm 4.9}  \newline
> \hline
> \end{array}
>
> ---
>
> **[C3] How is the $\Delta t$ handled during the training?**
>
> In all experiments, we sample $\Delta t := |t_1 - t_2|$ by subtracting the values from  two different beta distributions $t_1 \sim \mathtt{Beta}(2,1)$, $t_2 \sim \mathtt{Beta}(1,2)$ for both real and generated videos. Such distributions can sample $\Delta t$ diversely (e.g., measured with 10,000 samples, 36.5\%, 42.7\%, and 20.8\% of $\Delta t$ is in the interval of $[0,\frac{1}{3}]$, $[\frac{1}{3}, \frac{2}{3}]$, and $[\frac{2}{3}, 1]$, respectively), and indeed worked well in our experiments. We also update Section 3.2 and Appendix A to include this comment in the revision.
>
> ---
>
> **[C4] More results on the effect of diverse motion vectors.**
>
> Following your suggestion, we provide more comparison of videos with the different motion vectors in our project page. The results on diverse datasets confirm that different motion vectors generate various videos conditioned on the same contents, i.e., identical initial frames.
>
> ---
>
> **[C5] Why is only TaiChi used for the interpolation task?**
>
> We remark that we set the stride as 4 for the TaiChi dataset to make the motion more dynamic, while stride 1 was considered for other datasets. Accordingly, we only provided quantitative interpolation results on TaiChi in Table 2 since ground-truth interpolated frames are accessible only for this dataset (but we also provided the qualitative interpolation result on the Sky dataset on the project page). Nevertheless, following your suggestion, we provide additional interpolation results in the below table by training both our method and MoCoGAN-HD with Sky and Kinetics datasets of stride 4 as we did for TaiChi. The result (FVD, lower is better) confirms the superiority of our method on interpolation on various datasets. We also updated Table 2 to include this result in the revision and updated the project page to include more qualitative results.
>
> \begin{array}{l cc}
> \hline
>   & \text{Sky} & \text{Kinetics-food} \newline
> \hline
> \text{MoCoGAN-HD} & 402.2\small{\pm18.9} & {1029.8} \small{\pm28.4} \newline
> \text{DIGAN (ours)} & \mathbf{324.2\small{\pm20.5}} & \mathbf{{722.2} \small{\pm20.1}} \newline
> \hline
> \end{array}

---

> ### Author Response · Authors · 2021-11-22
> **A gentle reminder**
>
> Dear Reviewer Tj68,
>
> Thank you for your time and efforts in reviewing our paper.
>
> We kindly remind that the discussion period will end in 30 hours or so.
>
> We sincerely hope that our response and results of the supporting experiments successfully clarify your concerns.
>
> We just wonder whether we could have the last chance to address your further concerns or questions (if you have any).
>
> Thank you very much!
>
> Authors

---

### Official Review · Reviewer_ErkP · 2021-11-02

**Correctness:** 4
**Technical Novelty And Significance:** 3
**Empirical Novelty And Significance:** 3
**Recommendation:** 6
**Confidence:** 4

**Main Review:**

The paper presents a good submission, is well written, well motivated and evaluated.

Strengths:
1. Idea of representing videos as continuous functions. I believe this is an important idea not addressed in the past
2. The idea of temporally conditioned video discriminator. A heavy weight discriminator is a typical problem in video synthesis, as it makes training of the whole pipeline longer. I wonder if a similar discriminator can be used to make previous models be able to generate longer sequences.
3. Results and applications. The reported results show significant numerical improvements over the state-of-the-art methods. What's more interesting, similar to other INR methods, the present work can extrapolate videos in space and time. It can also do video in beetwening.

Weaknesses:
1. The generator part is adaptation of the INR-GAN to the temporal domain. The extension is simple.
2. The paper doesn't discuss the limitations. I wonder if the method can be trained on extremely long sequences, or perhaps can generate a couple of tens of seconds at inference time? The provided website contain a limited of 2 seconds long examples and two more 4 seconds long examples, which look like interpolation.
3. Consider two frames of a video, for example, when the hand is up and the other one when the hand is down. Since the action can be performed fast (or slow), there can be arbitrary many (or few) frames between these two frames. It's not clear if this adds any difficulties for the video discriminator, since the conditioning can be seen as incorrect. Could the authors comment?
4. I wonder if the authors thought of any metric that show that the proposed motion is motion indeed, rather than interpolation between two or more frames. For example, in Fig. 12 the generated results look like interpolations between two frames. See the example on the top, the right hand grows, not moves. One could have an experiment, in which indeed an interpolation in StyleGAN space is made and then compared to these results.

UPDATE:

I read the authors' response. It seems to me, that both the generator and discriminator work because of the relative simplicity of commonly used benchmarks for video synthesis. It means that a dataset can be well described by a pair of frames sampled many times from the data.   UCF-101 is a way more challenging dataset (too challenging) for current video synthesis methods. This is, however, not an issue of this paper but of the field in general.

It seems odd to me, that authors showed only 2 examples of long video generation on TaiChi (where the method seems to work well) and 100 examples on UCF-101 where the method works very poorly, generates repetitive motions, generates videos that don't make sense (yet better than state-of-the-art). How representative are the two examples? I had the same question in my initial review. The authors had two 4-second examples, now they have 2 8-second example which doesn't bring much difference.

So I believe my initial evaluation was correct, the paper is slightly more on the positive side of the bar.

**Summary Of The Paper:**

The paper present a new method to video generation. They built on top of the INR-GAN framework and extend it to temporal domain. They, therefore, get continuous time and space image generation--something all other video methods lack (technically one should be able to interpolate motion codes of existing CNN-based works, but spatially none of them is continuous to my knowledge). They model a video as a function $f(x,y,t)$. They condition this functions on motion codes and on content codes. The upper layers get information from the content, the lower layers get information from both content and motion. They further introduce a discriminator that is conditioned on the time difference between frames. It allows them to train on longer videos.


**Summary Of The Review:**

I believe, the paper is a good piece of work and is above the bar.

---

> ### Author Response · Authors · 2021-11-16
> **Response to Reviewer ErkP (2/2)**
>
> **[C4] Quantitative comparison to StyleGAN interpolation on video prediction.**
>
> As you suggested, we report structural similarity index measure (SSIM) to quantitatively compare predicted videos from DIGAN and linearly interpolated image sequences in StyleGAN2 [2] latent space. For StyleGAN2, we use the official StyleGAN2 inversion method, namely, we project each of the initial and final frames to the StyleGAN2 latent space trained on the TaiChi dataset. After that, we linearly interpolate on these two projected latent vectors. As shown in the below table, DIGAN shows better prediction than StyleGAN2 interpolation. We also update the qualitative comparison in Appendix E.2; one can observe that DIGAN preserves semantics like background across the temporal direction, while the StyleGAN2 latent interpolation cannot.
>
> \begin{array}{lc}
> \hline
>  & \text{SSIM }(\uparrow)  \newline
> \hline
> \text{StyleGAN2 latent space interpolation} & 0.5639 \newline
> \text{DIGAN (ours)} & \mathbf{0.6753} \newline
> \hline
> \end{array}
>
> ---
>
> [1] Skorokhodov et al., Adversarial Generation of Continuous Images, CVPR 2021\
> [2] Karras et al., Analyzing and Improving the Image Quality of StyleGAN, CVPR 2020

---

> ### Author Response · Authors · 2021-11-16
> **Response to Reviewer ErkP (1/2)**
>
> We sincerely appreciate your efforts and insightful comments to improve the manuscript. We respond to each of your comments one-by-one in what follows. In the revised draft, we mark our major revisions by “blue”. We also provide additional examples of videos in our project page: https://iclr2022-digan.github.io.
>
> ---
>
> **[Q1] Can the proposed discriminator be used for training previous models?**
>
> Utilizing our discriminator in previous models is also possible. However, it is not clear how to train our discriminator efficiently under the previous models. This is because our discriminator requires a pair of images from two arbitrary timesteps, and previous models (which generate videos in an autoregressive manner or with convolutional networks) require the entire video generation to create a pair of images, in contrast to our method that can synthesize frames in parallel. Hence, a naive implementation of our discriminator under the previous models may suffer from computational inefficiency or poor sampling qualities. Nevertheless, we think utilizing our discriminator in previous models is an interesting direction to explore in the future.
>
> ---
>
> **[C1] Generator seems like a simple extension of INR-GAN.**
>
> As pointed out by Reviewer Tj68, we believe our method contains various novel (yet simple) components on the architecture designed by considering the temporal aspects of videos, e.g., small temporal frequency $\sigma_t$ and additional motion latent vector $z_M$. Moreover, in the below table (i.e., Table 5 in the manuscript), we verify these components show considerable performance improvement from the simple extension of INR-GAN, i.e., they provide an improvement 686 $\rightarrow$ 577 in the FVD score, compared to the vanilla INR-GAN for video generation. Nevertheless, we believe the simplicity (with the effectiveness) of our method is rather the strength of our work, in particular, because we tried a completely new paradigm (based on INR) for the task and following works might use ours as an important baseline in the future.
>
> \begin{array}{cccc}
> \hline
> \text{Freq. }  \sigma_{t}  & \text{Motion }  z_{M} & \text{MLP } f_{M}(\cdot) & \text{FVD }(\downarrow)  \newline
> \hline
> \text{-} & \text{-} & \text{-} & 686 \small{\pm 25} \newline
> \text{O} & \text{-} & \text{-} & 640 \small{\pm 22} \newline
> \text{O} & \text{O} & \text{-} & 585 \small{\pm 27} \newline
> \text{O} & \text{O} & \text{O} & 577 \small{\pm 21} \newline
> \hline
> \end{array}
>
> ---
>
> **[C2] Synthesis of extremely long videos and discussion on the limitations.**
>
> To address your question, on the project page, we provide 256 frame videos generated by our model, trained with the TaiChi dataset consisting of 256 frame videos (they are 208 frames longer than the previous state-of-the-art method of 48 frames). Even though the video length becomes much longer than the previous state-of-the-art method, our model still generates valid Taichi motions while preserving the identity of the martial artist. Here, we remove 32.9\% of video clips shorter than 256 frames, and the training is underfitted due to the limited time; hence, the video quality of our method can be further improved if the dataset size becomes larger and more training iterations are performed.
>
> Even though our method is capable of generating long videos, we think it still remains a challenge to synthesize videos that contain discontinuous scene changes, i.e., two different videos are concatenated. Developing the method for synthesizing these videos should be an interesting future direction, and we believe our approach to incorporating INRs for video generation can provide new insights on building them.
>
> ---
>
> **[C3] Can our discriminator handle various possibilities of action speed?**
>
> Our discriminator can handle various possibilities of action speeds, and all feasible time differences of image pairs that are in the training dataset should be considered “correct”. As you pointed, the given two frames may have multiple feasible time differences, smaller for fast motions and larger for slow motions. Since our discriminator knows that both cases are real, our generator can create videos of both fast and slow motions (yet not create infeasible motions). For instance, our method trained on the Sky dataset (which includes different time-lapse speeds) generates sky videos of different rates, as shown in the main result on the project page.

---

> ### Author Response · Authors · 2021-11-22
> **Response to Reviewer ErkP's update**
>
> Dear Reviewer ErkP,
>
> We are truly grateful for taking your time to provide additional recommendations and acknowledge our efforts.
>
> We just reported two examples of long videos on TaiChi to visualize better (and emphasize) long video examples; nevertheless, the overall trend in quality on other generated videos is similar. To confirm this, in our project page (https://iclr2022-digan.github.io), we added 100 randomly sampled (i.e., not cherry-picked) videos of 256 frames generated by our method on TaiChi. As you mentioned, our method works well on the TaiChi dataset, generating considerable motion changes while preserving the person's identity like the two representative examples previously reported. We remark that our updated 8-second examples are twice longer than the 4-second ones (we considered in our initial submission), which may look seemingly similar, but technically much more challenging since it requires modeling twice longer, dynamic motion per video at once. We also emphasize that this is a significant improvement over the previous state-of-the-art method [1] that synthesized 48 frame videos (while we synthesized 5.3 times longer 256 frame videos).
>
> Moreover, as you mentioned, UCF-101 is a much more challenging dataset and all current video synthesis methods fail to create realistic videos. As you agreed, our method is better than the current state-of-the-art; but we also admit that there is still a gap from real videos. Nevertheless, we believe (or hope) that our work introducing a new paradigm of INR-based video generation would be an important step towards creating more realistic and complex videos.
>
> We sincerely hope that this result successfully clarifies your concerns.
>
> Authors
>
> ---
>
> [1] Clark et al., Adversarial Video Generation on Complex Datasets, 2019

---

### Official Review · Reviewer_3svg · 2021-11-09

**Correctness:** 3
**Technical Novelty And Significance:** 2
**Empirical Novelty And Significance:** 2
**Recommendation:** 5
**Confidence:** 4

**Main Review:**

The paper is well-written, easy to follow, and provides experiments on multiple datasets.
#### Strengths
- The paper provides a comparison to recent methods on IS, FVD, and KVD evaluation metrics. This helps to understand better the performance of the introduced model on video generation.
- Introducing an INR-based design for long-term video generation seems a good idea.
- Non-autoregressive generation. This is the feature I like the most. Being able to generate arbitrary time or predict past or future frames.

#### Weaknesses
 - Decomposing signal into motion and content is interesting however this is not new and previously studied by MoCoGAN[1] and Temporal Shift GAN [2] (which is not cited in the paper). We don't see any discussion in the related works on decomposition and similarity between works.
- Although extending INR-based image generator to video generator is interesting, it's a trivial modification and not significantly novel.
-  The paper states that "the proposed generator encourages the temporal coherency of videos by regulating the
variations of motion features and enhancing the expressive power of motions with an extra nonlinear mapping", however, this is not very clear that how is this done in the design (explicitly or implicitly)? Is there any specific term to control the temporal coherency of video in the generator? How do you "regulate variations of motion features"? I couldn't find the details in the paper.
- There is no information on computational complexity, training time, and the number of GPUs used for training.
- I really don't think that the discriminator design works for long-term video generation. Without seeing a good amount of video frames how do you expect the discriminator to evaluate the action? let's assume that in the long-term video a person doing something in the first 4 seconds and then goes out of the scene and another person starts doing something else! Then only two frames and temporal difference can't help the generator to create better motions. I think this discriminator works if the video is short and there is no repetition or high similarity in the video frames. To evaluate, a good experiment would be to generate random videos from UCF101 dataset and try to evaluate the performance with the metric introduced in [3] or a similar metric (S3) used in [2]. Or at least qualitatively show long videos from UCF101.
- Missing ref [4]



#### Ref:
1-MoCoGAN: Decomposing Motion and Content for Video Generation, Sergey Tulyakov, Ming-Yu Liu, Xiaodong Yang, Jan Kautz\
2-Temporal Shift GAN for Large Scale Video Generation, Andres Munoz, Mohammadreza Zolfaghari, Max Argus, Thomas Brox\
3-Classification Accuracy Score for Conditional Generative Models, Suman Ravuri, Oriol Vinyals\
4-StyleVideoGAN: A Temporal Generative Model using a Pretrained StyleGAN, Gereon Fox, Ayush Tewari, Mohamed Elgharib, Christian Theobalt

**Summary Of The Paper:**

The paper introduces an INR-based design that utilizes an MLP network to encode spatio-temporal dynamics of video. The authors also propose an efficient discriminator to detect unnatural motions. The design is an extension of INR-GAN (Skorokhodov et al., 2021) architecture. The paper claims that the method obtains SOTA performance on the UCF101 dataset.

**Summary Of The Review:**

Video generation is a very hard problem and the authors try to introduce a more efficient approach for this problem. I believe INR-based video generation is a valuable extension of the image generator but this is not significant. Also, the experiments cannot fully support the design choices for the discriminator.  Additionally, the paper misses some related works (and discussion on differences) and details on computational complexity and training time. I don't think the paper is ready to be accepted to ICLR. However, I'm open to discussions and will consider the author's response.

Update:
The author's response address almost all my questions, however, I'm not convinced about the discriminator choice and the author's explanation is not clear and convincing. This is also confirmed by their qualitative results on long videos such as the UCF101 video (64 frames). The result is much worse than DVD-GAN (Clark et al 2019).
I believe this work has positive aspects in terms of efficiency and ability to generate frames in parallel. Therefore, I'm increasing my rating from reject to "marginally below the acceptance threshold".

---

> ### Author Response · Authors · 2021-11-16
> **Response to Reviewer 3svg (3/3)**
>
> **[C5] Does the discriminator choice work for long-term video generation?**
>
> Our discriminator design (considering a triplet, i.e., pair of images and their time difference) works for long-term video generation by subsequently creating feasible short-term videos in the training dataset. Since the discriminator is trained on the samples of various time differences, it can evaluate if the action is feasible for the given time difference. Hence, the generator creates natural short-term videos for all pairs of close frames, leading to the long-term videos by inductively stacking them.
>
> The suggested scenario of large variation (person 1 goes out of the scene and person 2 comes in) is not problematic for our discriminator. Although the person in the scene has changed, the pair of frames from those scenes (i.e., one of person 1 and one of person 2) should be considered as real if the time difference between them is large. Since this large variation does not happen within the small time differences in the real videos, the generator does not create such large scene changes for the adjacent frames. In contrast, the action of each person can be learned from short-term videos between the pair of close frames. The repetition of video frames is not also problematic; the generator creates non-stationary videos if not all adjacent frames are stopped in the real videos. Here, the repeated video frames guide the generated frames back to the repeated frame after long time differences.
>
> For empirical validation, we additionally provide long video generation results on Tai-Chi-HD and UCF-101 datasets. First, recall that we already provided 128 frame Tai-Chi-HD videos, synthesized by DIGAN trained on 128 frame Tai-Chi-HD videos, in Figure 1. We remark that it is already 80 frames longer than the 48 frames of the previous state-of-the-art method [9]. To verify that DIGAN can handle even longer sequences, we added 256 frame Tai-Chi-HD videos, synthesized by DIGAN trained on 256 frame Tai-Chi-HD videos, on our project page. DIGAN subsequently creates valid Tai-Chi motions while preserving the identity of the martial artist.
>
> Following your suggestion, we also provide long video results on UCF-101. Specifically, we added 64 frame UCF-101 videos, synthesized by DIGAN trained on 64 frame UCF-101 videos, on our project page. Note that it is 4 times longer than the common benchmark setup of 16 frames (we choose 64 frames since many videos in the UCF-101 dataset are shorter or equal than 64 frames). The generated videos produce a longer variation of motions while preserving the temporal coherency.
>
> Finally, we recall that our focus is unconditional video GANs as in [1,3], and the suggested classifier-based metrics [2,6] are not suitable for our model as they are designed for class-conditional GANs. Nevertheless, we think the class-conditional extension of DIGAN would also be competitive for those metrics, as DIGAN creates diverse motions as qualitatively shown on the project page.
>
> ---
>
> **[C6] Add reference.**
>
> Thank you for pointing this out. We added [7] in the revision; see the discussion on video generation in Section 2. We also added [2] here.
>
> ---
>
> [1] Tulyakov et al., MoCoGAN: Decomposing Motion and Content for Video Generation, CVPR 2018\
> [2] Munoz et al., Temporal Shift GAN for Large Scale Video Generation, WACV 2021\
> [3] Tian et al., A Good Image Generator Is What You Need for High-resolution Video Synthesis, ICLR 2021\
> [4] Villegas et al., Decomposing Motion and Content for Natural Video Sequence Prediction, ICLR 2017\
> [5] Hsieh et al., Learning to Decompose and Disentangle Representations for Video prediction, NeurIPS 2018\
> [6] Ravuri et al., Classification Accuracy Score for Conditional Generative Models, NeurIPS 2019\
> [7] Fox et al., StyleVideoGAN: A Temporal Generative Model using a Pretrained StyleGAN, 2021\
> [8] Liu et al., Content-Aware GAN Compression, CVPR 2021\
> [9] Clark et al., Adversarial Video Generation on Complex Datasets, 2019

---

> ### Author Response · Authors · 2021-11-16
> **Response to Reviewer 3svg (2/3)**
>
> **[C4] Information on computational complexity.**
>
> Thank you for pointing out this issue. All the experiments are processed with 4 NVIDIA V100 32GB GPUs where it takes at most $\sim$4.4 days to complete. In addition, we provide the analysis of our method on efficiencies of diverse aspects, including computation, energy, memory, and time efficiency (we add Appendix I in the revision). Here, we mainly compared our method with MoCoGAN-HD [3], which is an energy-efficient (i.e., shorter GPU days for training) and state-of-the-art method on the generation quality. All experiments are performed under the same machine (NVIDIA V100 32GB GPUs).
>
> - **Computational efficiency.** We report floating point operations per second (FLOPS) of video GAN generators (following [8]). The DIGAN generator requires 147.9 GFLOPS to generate a 16 frame video of 128$\times$128 resolution, 4.6 times smaller than 682.3 GFLOPS of the MoCoGAN-HD generator.
> - **Energy efficiency on training (GPU days for training).** The FVD value of DIGAN trained on UCF-101 for 8 GPU days achieves 689±24, which is better than 838 of MoCoGAN-HD trained for 16 GPU days. This implies that DIGAN is (at least 2 times) more energy-efficient than MoCoGAN-HD.
> - **Memory efficiency on training.** We report the memory size per GPU to load 16 frame videos of 128$\times$128 resolution of batch size 4 for training video GANs. DIGAN allocates 9.7GB memory, 2.9 times smaller than 28.0GB of MoCoGAN-HD.
> - **Time efficiency on inference.** Per each GPU, DIGAN discriminators can (forward) compute 195.6 video clips/sec, 3.2 times larger than 60.3 video clips/sec of MoCoGAN-HD discriminators with 128 frame videos of 128$\times$128 resolution and a batch size of 16. It confirms that both the generator and discriminator of our method are much more time-efficient; recall that we already demonstrated that our generator is ~2.3 times faster per each GPU in Table 3.
>
> In summary, DIGAN is more computation-efficient ($>$ 4.6 times), energy-efficient on training  ($>$ 2 times), memory-efficient on training ($>$ 2.9 times), and time-efficient on inference ($>$ 3.2 times) under each machine setup above. Such gaps stem from the fact that our method utilizes: (a) INR-based generator without time-consuming autoregressive modeling and (b) 2D convolutional discriminator rather than computationally heavy 3D convolutional discriminator. Finally, we emphasize that DIGAN can synthesize multiple frames of a video in parallel (which is impossible under prior autoregressive or convolutional network-based methods), resulting in $N$ times improved time-efficiency to generate a single video under $N$ number of GPUs. It can be particularly important for synthesizing extremely long videos; since it requires tremendous time if utilizing prior methods that are impossible to compute frames in parallel. Like this, the superiority of computational efficiency of DIGAN can be further dramatically improved in the presence of multiple GPUs.

---

> ### Author Response · Authors · 2021-11-16
> **Response to Reviewer 3svg (1/3)**
>
> We sincerely appreciate your efforts and insightful comments to improve the manuscript. We respond to each of your comments one-by-one in what follows. In the revised draft, we mark our major revisions by “blue”. We also provide additional examples of videos in our project page: https://iclr2022-digan.github.io.
>
> ---
>
> **[C1] Discussion on video decomposition into motion and content.**
>
> We first emphasize that our goal is to make video generation scalable by leveraging implicit neural representations (INRs), rather than a better decomposition of videos into motion and content. In this respect, we focus more on showing the effectiveness of INRs in video generation, yet we are the first to decompose videos into motion and content under the context of INR to improve our scheme. However, as you mentioned, we agree that such decomposition is widely studied under different architectures (e.g., CNN+LSTM) in the literature, which we think is definitely worth discussing for providing meaningful insights for readers.
>
> Accordingly, following your suggestion, we provide the related discussion, including prior approaches [1,2,3,4,5]. These approaches incorporate distinct motion and content encoders for a prediction [4,5] or generators for a generation [1,2,3] to decompose the motion and contents, where they represent the motion as a sequence of latent vectors. Similarly, we utilize two different generators for motion and content; but the critical difference in our work is that the motion is represented as a fixed-sized single vector (of the weight of INRs) rather than a (long) sequence of varying length. We also add this discussion to the revision (see Appendix C). Thank you very much for the suggestion !
>
> ---
>
> **[C2] Extension of INR-based generator seems not novel.**
>
> As pointed out by Reviewer Tj68, we believe our method contains various novel (yet simple) components on the architecture designed by considering the temporal aspects of videos, e.g., small temporal frequency $\sigma_t$ and additional motion latent vector $z_M$. Moreover, in the below table (i.e., Table 5 in the manuscript), we verify these components show considerable performance improvement from the simple extension of INR-GAN, i.e., they provide an improvement 686 $\rightarrow$ 577 in the FVD score, compared to the vanilla INR-GAN for video generation. Nevertheless, we believe the simplicity (with the effectiveness) of our method is rather the strength of our work, in particular, because we tried a completely new paradigm (based on INR) for the task and following works might use ours as an important baseline in the future.
>
> \begin{array}{cccc}
> \hline
> \text{Freq. }  \sigma_{t}  & \text{Motion }  z_{M} & \text{MLP } f_{M}(\cdot) & \text{FVD }(\downarrow)  \newline
> \hline
> \text{-} & \text{-} & \text{-} & 686 \small{\pm 25} \newline
> \text{O} & \text{-} & \text{-} & 640 \small{\pm 22} \newline
> \text{O} & \text{O} & \text{-} & 585 \small{\pm 27} \newline
> \text{O} & \text{O} & \text{O} & 577 \small{\pm 21} \newline
> \hline
> \end{array}
>
> ---
>
> **[C3] Temporal coherency by regulating the variation of motion features?**
>
> We remind you that our method interprets videos as INRs; this interpretation automatically achieves the temporal coherency of generated videos since INRs are continuous functions which enable smooth inter-/extra-polation across with time $t$. Moreover, we further regulate the variations of motion features by utilizing a smaller temporal frequency $\sigma_t$ than spatial frequencies $\sigma_x, \sigma_y$; which explicitly regularizes the temporal difference of videos not to be large (it is stated in Section 3.2). Nonetheless, we update Section 1 to explicitly state these components in the revision to make them more precise.

---

> ### Author Response · Authors · 2021-11-22
> **A gentle reminder**
>
> Dear Reviewer 3svg,
>
> Thank you for your time and efforts in reviewing our paper.
>
> We kindly remind that the discussion period will end in 30 hours or so.
>
> We sincerely hope that our response and results of the supporting experiments successfully clarify your concerns.
>
> We just wonder whether we could have the last chance to address your further concerns or questions (if you have any).
>
> Thank you very much!
>
> Authors

---

> ### Author Response · Authors · 2021-11-30
> **Response about the additional concern on comparison with DVD-GAN**
>
> We are happy to hear that our response could help to address your concerns well. Due to your valuable and constructive suggestions, we also believe that our paper is much improved.
>
> To our knowledge, DVD-GAN provides “class-conditional” generated video examples on UCF-101 for their qualitative results, while we primarily focus on “unconditional” generation throughout the paper. Thus, directly comparing the video qualities provided in our paper (and project page) and those in DVD-GAN is unfair. We remark that our method significantly outperforms DVD-GAN on the UCF-101 dataset (measured with IS metric) for unconditional video synthesis (see Table 1). Therefore, we strongly believe that the conditional version of our method would also outperform the conditional DVD-GAN. We will add both qualitative and quantitative results of “class-conditional” video generation of our method in the final manuscript.
>
> Thank you,\
> Authors

---

### Author Response · Authors · 2021-11-16
**General Response**

Dear reviewers and AC,

We sincerely appreciate your valuable time and effort spent reviewing our manuscript.

As reviewers highlighted, we believe our paper is well-motivated (all reviewers) and provides a novel (Tj68), effective (vhpn, ErkP, Tj68) method for video generation, while meriting various intriguing properties (all reviewers) and showing strong empirical results on extensive experiments (vhpn, Erkp, Tj68). In particular, we believe that our paper provides a meaningful step toward solving the challenging task of video generation, by proposing a completely new (yet simple) approach that achieves state-of-the-art results.

We appreciate your incisive comments on our manuscript. In response to the questions and concerns you raised, we have carefully revised and improved the manuscript with the following additional discussions and experiments:

- Clearer component-by-component description of the proposed DIGAN (Section 1, Section 3)
- Added references with additional discussion on motion/content decomposition from videos (Section 2, Appendix C)
- Additional experimental evaluation on DIGAN trained using only the training split of UCF-101 dataset (Table 1)
- More discussions and illustrations on video prediction (Figure 5, Appendix E)
- Quantitative results on video interpolation done on other datasets: Kinetics and Sky (Table 2)
- Clearer description of experiment/implementation details (Figure 9, Appendix A)
- Additional analysis on the effectiveness of components of DIGAN (Figure 10, Appendix J)
- Extensive analysis of our method on efficiency, including computation, energy, memory, and time (Appendix I)
- More examples of videos generated by DIGAN (project page)

These updates are temporarily highlighted in "blue” for your convenience to check.

We sincerely believe that DIGAN can be a useful addition to the ICLR community, in particular, due to the above updates helping us better deliver the effectiveness of our method.

Thank you very much !

Authors.

---

### Author Response · Authors · 2021-11-19
**A gentle reminder**

Dear Reviewers,

Thank you for your time and efforts in reviewing our paper.

We kindly remind that the discussion period will end soon (in a few days). We believe that we sincerely and successfully address your concerns/questions/misunderstandings/suggestions, with the results of the supporting experiments.

If you have any further concerns or questions, please do not hesitate to let us know.

Thank you very much!

Authors

---

### Decision · Program_Chairs · 2022-01-20

**Decision:**

Accept (Poster)

**Comment:**

This work tackles video generation using implicit representations, and demonstrates that using these representations enables improvements to long-term coherence of the generated videos.

Reviewers praised the writing, the thorough experimental evaluation, and the strong quantitative results. Some concerns were raised about a lack of discussion of relevant related work, novelty/significance, model architecture, and a lack of qualitative examples, many of which the authors have tried to address during the discussion phase. Several reviewers raised their ratings as a result.

Personally I certainly believe that exploring implicit representations for video is important, and I know of no published prior work in this direction, which amplifies the potential significance of this work. Even if results are qualitatively worse than previous work in some ways, this exploration is still valuable and worth publishing.

While the paper ultimately received one reject rating, another reviewer chose to champion this work and award it the highest possible rating. Combined with the other positive reviews, this provides plenty of convincing evidence for me to recommend acceptance. That said, given the rating spread, I would like to encourage the authors to consider the reviewers' comments further as they prepare the final version of the manuscript. Especially providing more qualitative results would be a welcome addition.